# EvoIR: Towards All-in-One Image Restoration via Evolutionary Frequency Modulation

## Abstract

All-in-One Image Restoration (AiOIR) tasks often involve diverse degradation that require robust and versatile strategies. However, most existing approaches typically lack explicit frequency modeling and rely on fixed or heuristic optimization schedules, which limit the generalization across heterogeneous degradation. To address these limitations, we propose EvoIR, an AiOIR-specific framework that introduces evolutionary frequency modulation for dynamic and adaptive image restoration. Specifically, EvoIR employs the Frequency-Modulated Module (FMM) that decomposes features into high- and low-frequency branches in an explicit manner and adaptively modulates them to enhance both structural fidelity and fine-grained details. Central to EvoIR, an Evolutionary Optimization Strategy (EOS) iteratively adjusts frequency-aware objectives through a population-based evolutionary process, dynamically balancing structural accuracy and perceptual fidelity. Its evolutionary guidance further mitigates gradient conflicts across degradation and accelerates convergence. By synergizing FMM and EOS, EvoIR yields greater improvements than using either component alone, underscoring their complementary roles. Extensive experiments on multiple benchmarks demonstrate that EvoIR outperforms state-of-the-art AiOIR methods.

## 1 Introduction

Image restoration recovers a high-quality image from its degraded observation. Traditionally, this problem has been tackled by task-specific networks, each tailored to a particular type of degradation. Such task-specific models have demonstrated impressive performance across various tasks, including denoising Shen et al. (2023), dehazing Song et al. (2023), deraining Chen et al. (2023a), deblurring Tsai et al. (2022), low-light enhancement Ma et al. (2023b), and under-water enhancement Zhang et al. (2025b).

However, task-specific methods suffer from limited generalization, as they are inherently tailored to handle only predefined degradation types. When applied to unfamiliar degradation, their performance tends to degrade dramatically. General image restoration approaches have been proposed Zamir et al. (2021); Chen et al. (2022); Cui et al. (2023a); Xia et al. (2023) to address these limitations. Although these models are capable of addressing various degradation types, they are generally trained and tested on single tasks, which limits their practicality in real-world settings that involve complex and mixed degradation.

Recently, All-in-One image restoration methods Ai et al. (2024); Conde et al. (2024); Liu et al. (2025); Cui et al. (2025); Zamfir et al. (2025); Tian et al. (2025) have emerged as promising solutions to the limitations aforementioned. These approaches restore images corrupted by multiple degradation types within a unified framework. Early efforts such as AirNet Li et al. (2022) constructed explicit degradation encoders to obtain discriminative degradation-aware features. Subsequent works, including ProRes Ma et al. (2023a) and PromptIR Potlapalli et al. (2023), enhanced performance by incorporating visual prompts as guidance. The work in Tan et al. (2024) exploits the rich feature representations of large-scale vision models, such as CLIP Radford et al. (2021) and DINO Caron et al. (2021). Perceive-IR Zhang et al. (2025a) formulates image restoration from a quality-aware perspective, enabling the model to adjust its restoration strategy based on degradation severity.

Despite the emergence of All-in-One restoration frameworks, two key challenges remain largely underexplored. First, most existing methods operate solely in the spatial domain and fail to explicitly model the frequency characteristics of degraded images. This limitation hinders the ability of the model to balance the restoration of structural smoothness and texture fidelity. Second, current training strategies are typically static, relying on fixed loss weights throughout optimization. Such rigid configurations prevent the model from dynamically adapting to the varying difficulty levels across samples or tasks. These limitations lead to suboptimal performance in complex degradation scenarios.

To address these limitations, we propose **EvoIR**, an All-in-One image restoration framework that integrates frequency-aware representation learning with dynamic loss optimization. At its core, we introduce a **Frequency-Modulated Module (FMM)**, which decomposes features into high- and low-frequency components and applies branch-specific modulation to enhance texture details and preserve structural consistency. Additionally, we develop an **Evolutionary Optimization Strategy (EOS)** that simulates population-based evolution during training to dynamically adjust loss weight configurations, allowing the model to adapt to varying restoration objectives and task complexities without manual tuning.

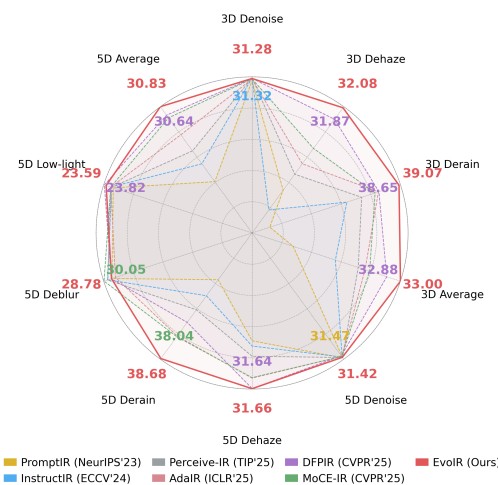

Figure 1: PSNR comparisons of AiOIR methods on "Noise + Haze + Rain" (3D) and "Noise + Haze + Rain + Blur + Low-light" (5D) settings. Results of our EvoIR are marked in Red, while the other best results are indicated in each color. EvoIR performs the best in average.

As shown in Fig. 1, EvoIR consistently outperforms previous State-Of-The-Art (SOTA) All-in-One Image Restoration (AiOIR) methods across a wide range of degradation types. Especially, EvoIR achieve new highest results of average PSNR/SSIM in both 3-task and 5-task settings compared with recent proposed methods.

Our contributions can be summarized as follows:

- We propose **EvoIR**; to the best of our knowledge within AiOIR, it is the first framework that leverages an evolutionary algorithm for loss weighting, together with frequency-aware modulation. EvoIR attains state-of-the-art performance across multiple benchmarks and remains robust to diverse degradation.

- We introduce a **Frequency-Modulated Module (FMM)** that explicitly separates features into high- and low-frequency components and dynamically modulates each branch to target fine-grained textures and structural smoothness under complex degradation.

- We present an **Evolutionary Optimization Strategy (EOS)**, a population-based mechanism with modest overhead that automatically identifies and adapts optimal loss-weight configurations for AiOIR, improving convergence and balancing perceptual quality without manual tuning.

## 2 RELATED WORK

### 2.1 ALL-IN-ONE IMAGE RESTORATION

All-in-One image restoration methods address diverse degradation using a unified model, offering improved storage and deployment efficiency over task-specific Yasarla & Patel (2019); Chen et al. (2023a); Kupyn et al. (2018); Cai et al. (2023); Wu et al. (2022) and general-purpose Chen et al. (2022); Zamir et al. (2022b); Xia et al. (2023); Guo et al. (2024a) approaches.

The core challenge lies in restoring multiple degradation types within a shared parameter space. To address this, AirNet Li et al. (2022) applies contrastive learning for degradation discrimination, IDR Zhang et al. (2023) adopts a two-stage ingredient-oriented design, and ProRes Ma et al. (2023a) introduce visual prompts for guided restoration. Recent methods Tan et al. (2024); Luo et al. (2023) further leverage large-scale pre-trained vision models to enhance texture and semantics.

However, most approaches utilize similar restoration strategies across spatial regions, neglecting frequency properties and structural complexity, which leads to oversmoothing or texture loss. Moreover, fixed training objectives hinder adaptation to sample difficulty. EvoIR addresses these limitations through frequency-aware modulation and adaptive optimization for more robust restoration.

## 2.2 FREQUENCY DOMAIN-BASED IMAGE RESTORATION

Recent studies have emphasized the importance of frequency-domain modeling in boosting restoration performance Cui et al. (2023b); Wu et al. (2025). CSNet Cui et al. (2024) integrates channel-wise Fourier transforms and multi-scale spatial frequency modules, guided by frequency-aware loss, to enhance both spectral interaction and spatial detail. For deblurring, the Efficient Frequency Domain Transformer Kong et al. (2023) reformulates attention and feed-forward layers in the frequency domain, improving visual quality and efficiency. FPro Zhou et al. (2024) employs frequency decomposition and prompt learning to guide structure–detail recovery across diverse tasks. AdaIR Cui et al. (2025) further mines degradation-specific frequency priors and applies bidirectional modulation to enhance reconstruction.

While effective, these methods often exist solely and complex modules. In contrast, our adaptive frequency-modulated module that explicitly performs frequency-aware modulation in the feature space. With assist of evolutionary optimization, it balances texture–structure dynamically without incurring additional architectural overhead.

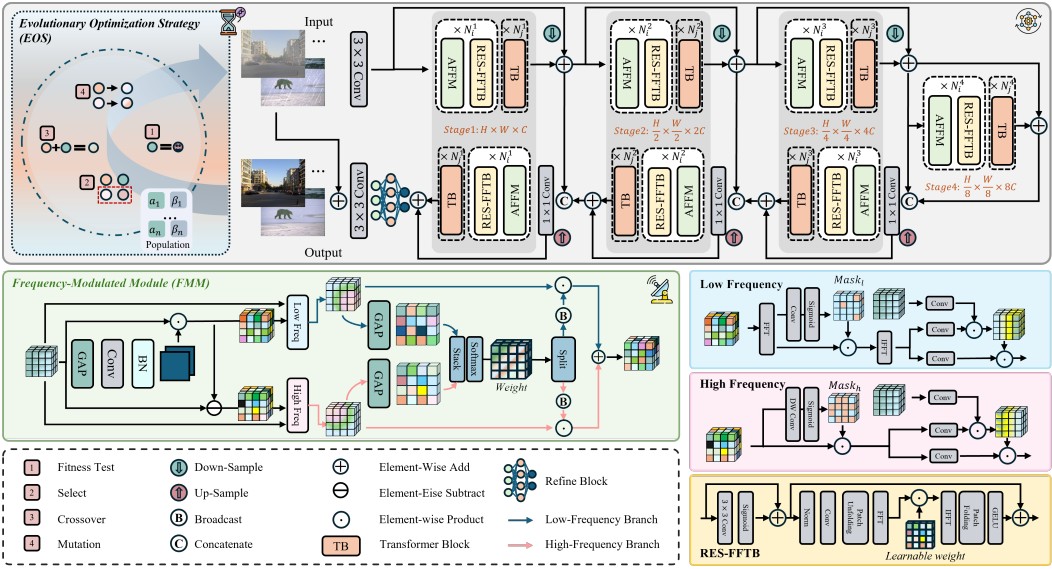

Figure 2: An overview of the EvoIR pipeline, combining frequency-aware representation (FMM), spectral-enhanced FFT blocks (RES-FFTB), and evolutionary loss optimization (EOS).

## 3 METHODOLOGY

### 3.1 OVERALL PIPELINE

As illustrated in Fig. 2, EvoIR comprises two tightly coupled components: a *frequency-modulated architecture* (FMM) and an *evolutionary optimization strategy* (EOS), which collaboratively achieve robust image restoration.

**Frequency-modulated architecture (FMM).** FMM splits the input features into a **low-frequency path** and a **high-frequency path** and fuses them adaptively. The low-frequency path performs *spectral gating*: it transforms the low-pass component to the frequency domain, applies a learned mask to suppress noise-dominated bands, and returns to the spatial domain for subsequent attention. The high-frequency path remains entirely in the *spatial domain*: a learned spatial mask emphasizes edges and textures, followed by depthwise convolutions to enhance fine-grained details. The two calibrated outputs are fused in the spatial domain and fed into multi-scale residual Transformer blocks (RES-FFTB) within a Restormer-like encoder–decoder, enabling hierarchical representation with strong long-range dependencies and local detail recovery.

**Evolutionary optimization (EOS).** To balance fidelity and perception under varying degradation without manual tuning, EOS performs a *stage-based* population search over loss weights $(\alpha, \beta) \in \Delta^2$. At the beginning of each stage (every $T$ iterations), we freeze the current network parameters and evaluate candidates on a held-out validation set; elites are retained, offspring are generated via convex crossover and small mutation with projection back to the simplex, and the best pair $(\alpha_t^\star, \beta_t^\star)$ is selected after $G$ generations. The selected weights are then used for the next $K$ epochs of training. This procedure improves stability (evaluation under frozen weights) and tracks the moving optimum as the model evolves.

**Backbone architecture.** Following prior frequency-domain designs Kong et al. (2023), we extend them into the *RES-FFTB* module with multi-head self-attention and residual connections for efficient information flow. We also adopt the Refine Block from AdaIR Cui et al. (2025) to further polish the fused features before decoding.

To summarize, FMM provides content-adaptive frequency modulation (spectral for low-frequency, spatial for high-frequency), while EOS supplies data-driven loss balancing across stages. Together they form a unified, robust pipeline that generalizes across diverse restoration tasks and complex degradation.

## 3.2 Adaptive Frequency Modulation

To effectively handle diverse and spatially variant degradation in All-in-One image restoration, we propose the **Frequency-Modulated Module (FMM)**, explicitly integrating frequency-aware inductive bias into our model. As illustrated in the middle-left of Fig. 2, FMM employs a dual-branch structure designed to individually handle high- and low-frequency components of input feature maps. This dual-branch approach empowers the model to adaptively emphasize texture details or structural smoothness, guided by input degradation patterns.

Given an input feature map $\mathbf{X}_F \in \mathbb{R}^{H \times W \times C}$, we first obtain two *modulation features* by a spatial band-split:

$$\mathbf{X}_L = \mathbf{G}_L * \mathbf{X}_F, \qquad \mathbf{X}_H = \mathbf{X}_F - \mathbf{X}_L, \tag{1}$$

where $*$ denotes spatial convolution and $\mathbf{G}_L$ is a learnable (or parametrized) low-pass kernel (e.g., depthwise separable). For interpretation, we also denote the unitary 2D Fourier transform by $\mathcal{F}$ and index Fourier coefficients by $\xi = (u, v)$ on the $H \times W$ DFT grid; then Eq. equation 1 is equivalent to $U_L(\xi) = \widehat{G}_L(\xi) U_F(\xi)$ and $U_H(\xi) = (1 - \widehat{G}_L(\xi)) U_F(\xi)$ in the frequency domain, but the high-frequency branch does not perform FFT/IFFT in implementation.

**Low-frequency branch (spectral gating).** We transform $\mathbf{X}_L$ to the frequency domain and apply a data-adaptive spectral mask $\mathbf{Mask}_\mathrm{l}(\xi) \in [0, 1]$:

$$U_L = \mathcal{F}(\mathbf{X}_L), \quad \widetilde{U}_L(\xi) = \mathbf{Mask}_\mathrm{l}(\xi) U_L(\xi), \quad \widetilde{\mathbf{X}}_L = \mathcal{F}^{-1}(\widetilde{U}_L). \tag{2}$$

The refined low-frequency features $\widetilde{\mathbf{X}}_L$ serve as *Key/Value*, while the original $\mathbf{X}_F$ acts as *Query* in the subsequent attention, enhancing structural smoothness and global coherence.

**High-frequency branch (spatial gating).** In parallel, we keep the computation entirely in the spatial domain and generate a spatial mask $\mathbf{m}_\mathrm{h} \in [0, 1]^{H \times W}$ conditioned on the input (via the global token from GAP and lightweight layers), and apply

$$\widetilde{\mathbf{X}}_H = \mathbf{m}_\mathrm{h} \odot \mathbf{X}_H, \tag{3}$$

followed by depthwise convolutions to emphasize fine-grained textures and edges. Especially, both FFT and IFFT are removed to better extract high-frequency information.

**Fusion.** Since the low branch returns to the spatial domain in Eq. equation 2, we fuse the two paths in the spatial domain:

$$\widehat{\mathbf{X}} = \widetilde{\mathbf{X}}_L + \widetilde{\mathbf{X}}_H. \tag{4}$$

The calibrated features are then processed by residual Transformer blocks (RES-FFTB), strengthening the decoder pathway and improving representation fidelity.

Finally, the calibrated features are processed through residual Transformer Blocks (TBs), specifically enhanced by the frequency-aware RES-FFTB module (as illustrated in the bottom-right of Fig. 2). RES-FFTB leverages learnable weights to modulate frequency-transformed features, further strengthening the decoder pathway and improving feature representation fidelity.

---

**Algorithm 1:** Evolutionary Optimization Strategy (EOS)

---

**Input:** Validation set $\mathcal{D}_v$; losses $\mathcal{L}_{\text{fid}}, \mathcal{L}_{\text{perc}}$; population size $n$; generations $G$; trigger interval $T$
      (iterations)
**Output:** Weights $(\alpha_r^\star, \beta_r^\star)$ applied for the next $T$ iterations
1   **Trigger $r$:** (called every $T$ training iterations) freeze current weights $\theta_r$ ;
2   Initialize population $\mathcal{P}_0 = \{(\alpha_i, \beta_i)\}_{i=1}^n$ with $\alpha_i + \beta_i = 1$ ;
3   **for** $g \leftarrow 1$ **to** $G$ **do**
4      **foreach** $(\alpha, \beta) \in \mathcal{P}_{g-1}$ **do**
         `// validation fitness under frozen` $\theta_r$
5          $f(\alpha, \beta) \leftarrow -\frac{1}{|\mathcal{D}_v|} \sum_{(x,y) \in \mathcal{D}_v} \left[ \alpha \, \mathcal{L}_{\text{fid}}(f_{\theta_r}(y), x) + \beta \, \mathcal{L}_{\text{perc}}(f_{\theta_r}(y), x) \right]$ ;
6      **end**
7      Keep top-$k$ elites $\mathcal{P}_{\text{elite}}$ by $f$ ;
8      Initialize new population $\mathcal{P}_g \leftarrow \mathcal{P}_{\text{elite}}$ ;
9      **while** $|\mathcal{P}_g| < n$ **do**
10         Sample parents $p_a, p_b \in \mathcal{P}_{\text{elite}}$ and $\lambda \sim \mathcal{U}(0, 1)$ ;
11         **Crossover:** $c = \lambda \, p_a + (1 - \lambda) \, p_b$ ;
12         **Mutation:** $c \leftarrow c + \varepsilon, \varepsilon \sim \mathcal{N}(0, \sigma^2)$ ;
13         **Projection:** $c \leftarrow \Pi_{\Delta^2}(c)$ ;      `// enforce` $\alpha + \beta = 1$ `and nonnegativity`
14         Add $c$ to $\mathcal{P}_g$ ;
15      **end**
16 **end**
17 **return** $(\alpha_r^\star, \beta_r^\star) = \arg\max_{(\alpha,\beta) \in \mathcal{P}_G} f(\alpha, \beta)$ ; `// use for the next` $T$ `iterations`

---

## 3.3 EVOLUTIONARY LOSS OPTIMIZATION

To enable adaptive balancing between fidelity and perceptual objectives under varying degradation severities, we introduce an **Evolutionary Optimization Strategy (EOS)** (Fig. 2, top-left). EOS treats the loss weights as candidates and performs a population-based search to find the best weight pair during training.

**Losses and feasible set.** We consider two batch-averaged losses: a pixel-wise fidelity loss $\mathcal{L}_{\text{fid}}$ (e.g., $\ell_1$/Charbonnier) and a perceptual loss $\mathcal{L}_{\text{perc}}$ (we use $1-$MS-SSIM so that a smaller value means better performance). A candidate is a weight pair $(\alpha, \beta) \in \Delta^2 = \{(\alpha, \beta) : \alpha, \beta \geq 0, \; \alpha + \beta = 1\}$. During training, the combined loss for a mini-batch is

$$\mathcal{L}_{\text{train}}(\theta; \alpha, \beta) = \alpha \, \mathcal{L}_{\text{fid}}(\theta) + \beta \, \mathcal{L}_{\text{perc}}(\theta), \qquad \alpha + \beta = 1. \tag{5}$$

**What EOS optimizes.** At trigger $r$ (every $T$ iterations), we *freeze* the current network parameters $\theta_r$ and evaluate each candidate on a held-out validation set $\mathcal{D}_v$:

$$J_r(\alpha, \beta \mid \theta_r) = \frac{1}{|\mathcal{D}_v|} \sum_{(x,y) \in \mathcal{D}_v} \left[ \alpha \, \mathcal{L}_{\text{fid}}(f_{\theta_r}(y), x) + \beta \, \mathcal{L}_{\text{perc}}(f_{\theta_r}(y), x) \right], \quad (\alpha, \beta) \in \Delta^2, \tag{6}$$

with fitness $f(\alpha, \beta) = -J_r(\alpha, \beta \mid \theta_r)$ (larger is better). EOS returns $(\alpha_r^\star, \beta_r^\star) = \arg\max f(\alpha, \beta)$ and uses it for the next $T$ training iterations by minimizing equation 5.

**Procedure.** As detailed in Alg. 1, EOS maintains a population $\mathcal{P}$ of weight pairs on the simplex. Each generation computes $f(\alpha, \beta)$ on $\mathcal{D}_v$ with $\theta_r$ frozen, keeps the top-$k$ *elites*, and creates offspring via convex *crossover* and small *mutation* followed by projection back to $\Delta^2$ (to keep $\alpha + \beta = 1$ and nonnegativity). This repeats for $G$ generations. The best $(\alpha_r^\star, \beta_r^\star)$ is then used for the next $T$ iterations until the next trigger. As detailed in Alg.1 and visualized in Fig.2, the EOS process involves the following steps:

1) **Initialization**: Randomly initialize (or pre-define) a small population $\mathcal{P}_0 = \{(\alpha_i, \beta_i)\}_{i=1}^n$ on $\Delta^2$.
2) **Fitness Evaluation (frozen $\theta_r$)**: For each $(\alpha, \beta) \in \mathcal{P}_{g-1}$, compute $f(\alpha, \beta) = -\frac{1}{|\mathcal{D}_v|} \sum_{(x,y) \in \mathcal{D}_v} \left[ \alpha \mathcal{L}_{\text{fid}}(f_{\theta_r}(y), x) + \beta \mathcal{L}_{\text{perc}}(f_{\theta_r}(y), x) \right]$.
3) **Selection (elitism)**: Keep the top-$k$ candidates by $f$ to form $\mathcal{P}_{\text{elite}}$.
4) **Crossover (convex)**: Sample parents $p_a, p_b \in \mathcal{P}_{\text{elite}}$ and $\lambda \sim \mathcal{U}(0, 1)$, set $c = \lambda\, p_a + (1 - \lambda)\, p_b$.
5) **Mutation + projection**: Perturb $c \leftarrow c + \varepsilon$ with $\varepsilon \sim \mathcal{N}(0, \sigma^2)$, then project $c \leftarrow \Pi_{\Delta^2}(c)$ to enforce $\alpha + \beta = 1$ and $c \geq 0$.
6) **Repeat & apply**: Iterate 2–5 for $G$ generations; return $(\alpha_r^\star, \beta_r^\star)$ and use it for the next $T$ training iterations.

With elitist selection and deterministic evaluation, the best fitness is non-decreasing across generations. In practice, we evaluate on a fixed stratified subset of $\mathcal{D}_v$ to reduce variance.

# 4 EXPERIMENTS

## 4.1 EXPERIMENTAL SETUP

**Datasets:** Following Li et al. (2022); Potlapalli et al. (2023), we consider *One-by-One training paradigm* (single-task training), *All-in-One training paradigm* (multi-task joint training), composited degradation and remote sensing imagery. For One-by-One and All-in-One, we explore two common degradation combinations: **N+H+R** (Noise, Haze, Rain) and **N+H+R+B+L** (Noise, Haze, Rain, Blur, Low-light). **Denoising**: BSD400 Arbelaez et al. (2010), WED Ma et al. (2016), CBSD68 Martin et al. (2001) and Kodak24 Franzen (1999); **Dehazing**: OTS from RESIDE-$\beta$ Li et al. (2018) and SOTS-Outdoor Li et al. (2018); **Deraining**: Rain100L Yang et al. (2017); **Deblurring**: GoPro Nah et al. (2017); **Low-light Enhancement**: LOL Wei et al. (2018). Details of these datasets are summarized in appendix. For composited degradation, we use CDD11 dataset Guo et al. (2024b). For remote sensing AiOIR, we choose MDRS-LandsatLihe et al. (2025). For these two settings, we follow the original split of training and test sets.

**Implementation Details:** For EvoIR, we employ the AdamW optimizer with $\beta_1 = 0.9$, $\beta_2 = 0.999$, and an initial learning rate of $2 \times 10^{-4}$. The model is trained for 150 epochs with a total batch size of 28. The loss weights are initialized as $\alpha = 0.8$ and $\beta = 0.2$, which are updated by the EOS. After 75 epochs, the learning rate is halved to $1 \times 10^{-4}$. The evolutionary optimization strategy is applied every 500 training iterations.

Following Cui et al. (2025), we adopt task-specific resampling ratios to address data imbalance across restoration tasks. Specifically, the data expansion ratios of 3, 120, 5, and 200 are applied to denoising, deraining, deblurring, and low-light enhancement, respectively, while dehazing remains unaltered. Training is conducted on 4 NVIDIA A100 40GB GPUs using cropped patches of size $128 \times 128$, with random horizontal and vertical flipping for data augmentation.

## 4.2 ALL-IN-ONE RESTORATION RESULTS

We evaluate EvoIR under two *All-in-One* settings: a moderate setting with three degradation and a more complex five-degradation setting .

**Three-Degradations Setting ("N+H+R").** Tab.1 shows that EvoIR achieves the best average PSNR (33.00 dB) and SSIM (0.922). Compared to the three latest AiOIR methods: AdaIR Cui et al. (2025), MoCE-IR Zamfir et al. (2025) and DFPIR Tian et al. (2025), our method improves the average PSNR/SSIM by +0.31 dB/+0.004, +0.23 dB/+0.005 and +0.12 dB/+0.003, respectively. Notably, EvoIR achieves new state-of-the-art results on dehazing (32.08/0.982) and deraining (39.07/0.985), demonstrating the effectiveness of adaptive frequency modulation and evolutionary optimization.

Table 1: Performance comparison (PSNR/SSIM) under All-in-One ("**N+H+R**") setting. Results are partially sourced from Perceive-IR Zhang et al. (2025a).

| Method | Denoising (CBSD68) | | | Dehazing | Deraining | Average | Params (M) |
|---|---|---|---|---|---|---|---|
| | $\sigma = 15$ | $\sigma = 25$ | $\sigma = 50$ | SOTS | Rain100L | | |
| AirNet (CVPR'22) | 33.92/0.932 | 31.26/0.888 | 28.00/0.797 | 27.94/0.962 | 34.90/0.967 | 31.20/0.910 | 8.93 |
| IDR (CVPR'23) | 33.89/0.931 | 31.32/0.884 | 28.04/0.798 | 29.87/0.970 | 36.03/0.971 | 31.83/0.911 | 15.34 |
| ProRes (ArXiv'23) | 32.10/0.907 | 30.18/0.863 | 27.58/0.779 | 28.38/0.938 | 33.68/0.954 | 30.38/0.888 | 370.63 |
| PromptIR (NeurIPS'23) | 33.98/0.933 | 31.31/0.888 | 28.06/0.799 | 30.58/0.974 | 36.37/0.972 | 32.06/0.913 | 32.96 |
| NDR (TIP'24) | 34.01/0.932 | 31.36/0.887 | 28.10/0.798 | 28.64/0.962 | 35.42/0.969 | 31.51/0.910 | 28.40 |
| Gridformer (IJCV'24) | 33.93/0.931 | 31.37/0.887 | 28.11/0.801 | 30.37/0.970 | 37.15/0.972 | 32.19/0.912 | 34.07 |
| InstructIR (ECCV'24) | **34.15**/0.933 | 31.52/0.890 | 28.30/0.804 | 30.22/0.959 | 37.98/0.978 | 32.43/0.913 | 15.84 |
| Up-Restorer (AAAI'25) | 33.99/0.933 | 31.33/0.888 | 28.07/0.799 | 30.68/0.977 | 36.74/0.978 | 32.16/0.915 | 28.01 |
| Perceive-IR (TIP'25) | 34.13/0.934 | **31.53**/0.890 | **28.31**/0.804 | 30.87/0.975 | 38.29/0.980 | 32.63/0.917 | 42.02 |
| AdaIR (ICLR'25) | 34.12/0.935 | 31.45/0.892 | 28.19/0.802 | 31.06/0.980 | 38.64/0.983 | 32.69/0.918 | 28.77 |
| MoCE-IR (CVPR'25) | 34.11/0.932 | 31.45/0.888 | 28.18/0.800 | 31.34/0.979 | 38.57/0.984 | 32.73/0.917 | 25.35 |
| DFPIR (CVPR'25) | 34.14/0.935 | 31.47/0.893 | 28.25/0.806 | 31.87/0.980 | 38.65/0.982 | 32.88/0.919 | 31.10 |
| **EvoIR** | 34.14/**0.937** | 31.48/**0.896** | 28.23/**0.811** | **32.08/0.982** | **39.07/0.985** | **33.00/0.922** | 36.68 |

As illustrated in Fig.3, EvoIR significantly enhances texture clarity and preserves structural details. In particular, for local red regions in the first line for denoising comparison, our method effectively restores fine local textures better than DFPIR and MoCE-IR, significantly improving visual clarity and realism. Structural elements, including edges and object boundaries, are distinctly sharper and more coherent compared to baseline methods, indicating EvoIR's superior capability in handling spatially variant degradation.

**Five-Degradations Setting ("N+H+R+B+L").** Tab. 2 shows the performance under a more challenging scenario. EvoIR maintains superior performance with an average PSNR of 30.83 dB and SSIM of 0.918. It still surpasses several recent methods like Perceive-IR, AdaIR, and DFPIR in both PSNR and SSIM. Even as the degradation types increase, EvoIR demonstrates remarkable stability, outperforming all SOTAs in PSNR and the second average SSIM (marginally -0.001).

Considering that our EvoIR has 36.68M parameters, it is on the same scale as other methods with better results. These consistent improvements validate the robustness and generalization capability of EvoIR under complex degradation scenarios.

### 4.3 ONE-BY-ONE RESTORATION RESULTS

We evaluate EvoIR under the *One-by-One* setting, where each restoration task is trained and tested independently. Top lines are task-specific and general methods, and the bottom lines refer to AiOIR methods.

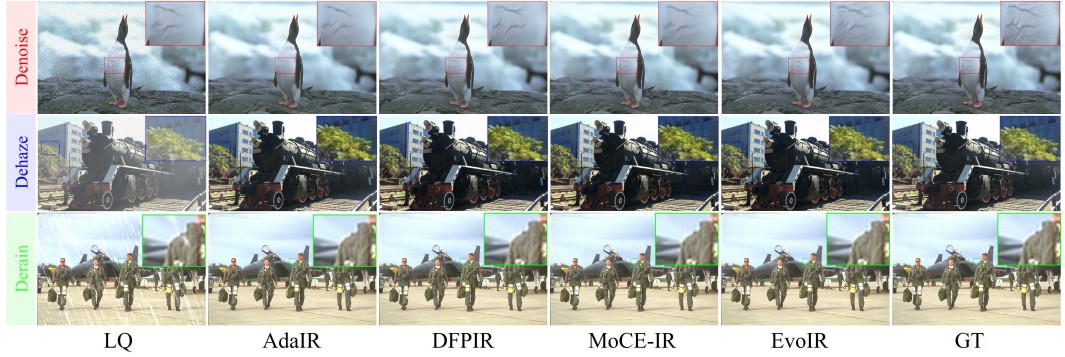

| LQ | AdaIR | DFPIR | MoCE-IR | EvoIR | GT |

Figure 3: Visual comparisons of EvoIR with state-of-the-art All-in-One methods under "**N+H+R**" setting. More visualization results are illustrated in appendix.

Table 2: Performance comparison (PSNR/SSIM) under All-in-One ("**N+H+R+B+L**") setting. Denoising results only reports noise level under $\sigma = 25$ following Zhang et al. (2023). Results are partially sourced from Perceive-IR Zhang et al. (2025a).

| Method | Denoising CBSD68 | Dehazing SOTS | Deraining Rain100L | Deblurring GoPro | Low-light LOL | Average | Params (M) |
|---|---|---|---|---|---|---|---|
| TAPE (ECCV'22) | 30.18/0.855 | 22.16/0.861 | 29.67/0.904 | 24.47/0.763 | 18.97/0.621 | 25.09/0.801 | 1.07 |
| Transweather (CVPR'22) | 29.00/0.841 | 21.32/0.885 | 29.43/0.905 | 25.12/0.757 | 21.21/0.792 | 25.22/0.836 | 37.93 |
| AirNet (CVPR'22) | 30.91/0.882 | 21.04/0.884 | 32.98/0.951 | 24.35/0.781 | 18.18/0.735 | 25.49/0.846 | 8.93 |
| IDR (CVPR'23) | **31.60**/0.887 | 25.24/0.943 | 35.63/0.965 | 27.87/0.846 | 21.34/0.826 | 28.34/0.893 | 15.34 |
| PromptIR (NeurIPS'23) | 31.47/0.886 | 26.54/0.949 | 36.37/0.970 | 28.71/0.881 | 22.68/0.832 | 29.15/0.904 | 32.96 |
| Gridformer (IJCV'24) | 31.45/0.885 | 26.79/0.951 | 36.61/0.971 | 29.22/0.884 | 22.59/0.831 | 29.33/0.904 | 34.07 |
| InstructIR (ECCV'24) | 31.40/0.887 | 27.10/0.956 | 36.84/0.973 | 29.40/0.886 | 23.00/0.836 | 29.55/0.907 | 15.84 |
| Perceive-IR (TIP'25) | 31.44/0.887 | 28.19/0.964 | 37.25/0.977 | 29.46/0.886 | 22.88/0.833 | 29.84/0.909 | 42.02 |
| AdaIR (ICLR'25) | 31.35/0.889 | 30.53/0.978 | 38.02/0.981 | 28.12/0.858 | 23.00/0.845 | 30.20/0.910 | 28.77 |
| MoCE-IR (CVPR'25) | 31.34/0.887 | 30.48/0.974 | 38.04/0.982 | **30.05**/**0.899** | 23.00/0.852 | 30.58/**0.919** | 25.35 |
| DFPIR (CVPR'25) | 31.29/0.889 | 31.64/0.979 | 37.62/0.978 | 28.82/0.873 | **23.82**/0.843 | 30.64/0.913 | 31.10 |
| **EvoIR** | 31.42/**0.895** | **31.66**/**0.980** | **38.68**/**0.984** | 28.78/0.876 | 23.59/**0.855** | **30.83**/0.918 | 36.68 |

As shown in Tab. 3, EvoIR achieves the best denoising performance on Kodak24, consistently ranking first in all noise levels. Notably, it reaches 35.30 dB at $\sigma = 15$, surpassing Perceive-IR and Restormer by +0.46 dB and +0.52 dB, respectively. On the dehazing task, EvoIR leads on SOTS (32.21/0.984), outperforming DehazeFormer and other strong baselines without relying on task-specific priors. For deraining, it achieves 39.23/0.986 on Rain100L, better than the second best method Perceive-IR, highlighting its adaptability to fine-scale textures. HI-Diff and FSNet rank first and second in deblurring, while all AiOIR methods fail to deblur well in this setting. As most task-specific and general methods contain modules designed for blurry regions, it is suitable that EvoIR performs poorly. In low-light enhancement, EvoIR ranks the third in PSNR and first in SSIM (24.09/0.850), only trailing Retinexformer and MIRNet while outperforming all methods.

Overall, EvoIR consistently delivers leading or near-leading results across four single tasks except deblurring, confirming its robustness in diverse degradation scenarios.

## 4.4 COMPOSITE DEGRADATION RESULTS

Besides the standard 3D/5D protocols, we add a composite degradation evaluation on CDD covering single (L/H/R/S), all pairwise double, and representative triple mixes. Results are illustrated in Tab. 4. EvoIR obtains the best average (28.88 dB / 0.885), surpassing OneRestore (28.47/0.878, +0.41 dB / +0.007) and ranking 1st on 8/11 subsets (the rest 2nd). These results indicate that EvoIR maintains strong performance not only on single degradations but also under co-occurring artifacts, consistent with the design goal of frequency-aware modulation plus stage-wise training stability.

Table 3: Single-task restoration results, including denoising, dehazing, deraining, deblurring, and low-light enhancement. **Bold** denotes best overall; underline highlights the second.

| Denoising Kodak24 | PSNR | | | Dehazing SOTS | PSNR/SSIM | Deraining Rain100L | PSNR/SSIM | Dreblurring GoPro | PSNR/SSIM | Low-light LOL | PSNR/SSIM |
|---|---|---|---|---|---|---|---|---|---|---|---|
| | $\sigma=15$ | $\sigma=25$ | $\sigma=50$ | | | | | | | | |
| DnCNN | 34.60 | 32.14 | 28.95 | DehazeNet | 22.46/0.851 | UMR | 32.39/0.921 | DeblurGAN | 28.70/0.858 | URetinex | 21.33/0.835 |
| FFDNet | 34.63 | 32.13 | 28.98 | FDGAN | 23.15/0.921 | LPNet | 33.61/0.958 | Stripformer | 33.08/0.962 | SMG | 23.81/0.809 |
| ADFNet | 34.77 | 32.22 | 29.06 | DehazeFormer | 31.78/0.977 | DRSformer | 38.14/0.983 | HI-Diff | **33.33**/**0.964** | Retinexformer | **25.16**/0.845 |
| MIRNet-v2 | 34.29 | 31.81 | 28.55 | Restormer | 30.87/0.969 | Restormer | 36.74/0.978 | MPRNet | 32.66/0.959 | MIRNet | 24.14/0.835 |
| Restormer | 34.78 | 32.37 | 29.08 | NAFNet | 30.98/0.970 | NAFNet | 36.63/0.977 | Restormer | 32.92/0.961 | Restormer | 22.43/0.823 |
| NAFNet | 34.27 | 31.80 | 28.62 | FSNet | 31.11/0.971 | FSNet | 37.27/0.980 | FSNet | 33.29/0.963 | DiffIR | 23.15/0.828 |
| AirNet | 34.81 | 32.44 | 29.10 | AirNet | 23.18/0.900 | AirNet | 34.90/0.977 | AirNet | 31.64/0.945 | AirNet | 21.52/0.832 |
| IDR | 34.78 | 32.42 | 29.13 | PromptIR | 31.31/0.973 | PromptIR | 37.04/0.979 | PromptIR | 32.41/0.956 | PromptIR | 22.97/0.834 |
| Perceive-IR | 34.84 | 32.50 | 29.16 | Perceive-IR | 31.65/0.977 | Perceive-IR | 38.41/0.984 | Perceive-IR | 32.83/0.960 | Perceive-IR | 23.79/0.841 |
| **EvoIR** | **35.30** | **32.86** | **29.78** | **EvoIR** | **32.21**/**0.984** | **EvoIR** | **39.23**/**0.986** | **EvoIR** | 29.57/0.891 | **EvoIR** | 24.09/**0.850** |

Table 4: Results of different image restoration methods under composite degradation images

| Methods | Single-L | Single-H | Single-R | Single-S | Double-L+H | Double-L+R | Double-L+S | Double-H+R | Double-H+S | Triple-L+H+R | Triple-L+H+S | Average |
|---|---|---|---|---|---|---|---|---|---|---|---|---|
| AirNet | 24.83/.778 | 24.21/.951 | 26.55/.891 | 26.79/.919 | 23.23/.779 | 22.82/.710 | 23.29/.723 | 22.21/.868 | 23.29/.901 | 21.80/.708 | 22.24/.725 | 23.75/.814 |
| PromptIR | 26.32/.805 | 26.10/.969 | 31.56/.946 | 31.53/.960 | 24.49/.789 | 25.05/.771 | 24.51/.761 | 24.54/.924 | 27.05/.925 | 23.74/.752 | 23.33/.747 | 25.90/.850 |
| WeatherDiff | 23.58/.763 | 21.99/.904 | 24.85/.885 | 24.80/.888 | 21.83/.756 | 22.69/.730 | 22.12/.707 | 21.25/.868 | 21.99/.868 | 21.23/.716 | 21.04/.698 | 22.49/.799 |
| WGSWNet | 24.39/.774 | 27.90/.982 | 33.15/.964 | 34.43/.973 | 24.27/.800 | 25.06/.772 | 24.60/.765 | 27.23/.955 | 27.65/.960 | 23.90/.772 | 23.97/.711 | 26.96/.863 |
| OneRestore | 26.48/.826 | **32.52**/.990 | 33.40/.964 | 34.31/.973 | 25.79/.822 | 25.58/.799 | 25.19/.789 | **29.99**/.957 | **30.21**/.964 | 24.78/.788 | 24.90/.791 | 28.47/.878 |
| EvoIR | **27.06/.830** | 32.24/**.991** | **34.03/.970** | **35.80/.981** | **25.92/.824** | **26.02/.806** | **25.96/.802** | 29.76/**.965** | 30.17/**.971** | **25.43/.797** | **25.31/.797** | **28.88/.885** |

## 4.5 REMOTE SENSING IMAGERY RESULTS

To better prove the effectiveness of our method, we include one new AiOIR task on remote sensing imagery. We evaluate on the recently proposed MDRS-Landsat all-in-one remote sensing benchmark. As is shown in Tab. 5, without any modification or special fine-tuning, EvoIR (37 M) outperforms prior SOTAs and even surpasses the remote-sensing-specialized Ada4DIR-d (41 M) across all four degradations.

Table 5: Results of different image restoration methods under remote sensing imagery

| Methods | Blur PSNR/SSIM | Dark PSNR/SSIM | Haze PSNR/SSIM | Noise PSNR/SSIM |
|---|---|---|---|---|
| NAFNet | 33.10/0.8120 | 30.40/0.9516 | 31.56/0.9642 | 33.08/0.8263 |
| Restormer | 35.23/0.8559 | 37.86/0.9872 | 36.18/0.9867 | 34.53/0.8589 |
| DGUNet | 29.64/0.7822 | 27.15/0.9010 | 27.45/0.9338 | 30.31/0.7314 |
| TransWeather | 33.45/0.8159 | 36.33/0.9705 | 35.02/0.9689 | 33.69/0.8428 |
| AirNet | 28.27/0.7887 | 28.38/0.9472 | 24.39/0.9331 | 30.30/0.7446 |
| PromptIR | 36.41/0.8861 | 39.09/0.9900 | 37.61/0.9897 | 34.99/0.8729 |
| IDR | 36.57/0.8902 | 35.19/0.9865 | 36.99/0.9892 | 34.88/0.8681 |
| SrResNet-AP | 34.63/0.8479 | 33.87/0.9823 | 34.78/0.9825 | 34.70/0.8620 |
| Restormer-AP | 35.75/0.8732 | 37.27/0.9885 | 37.36/0.9888 | 34.96/0.8697 |
| Uformer-AP | 34.64/0.8488 | 36.58/0.9899 | 36.06/0.9877 | 34.39/0.8533 |
| Ada4DIR-d | 37.20/0.9004 | 43.85/0.9954 | 41.06/0.9938 | 35.14/0.8724 |
| EvoIR | **37.48/0.9069** | **44.73/0.9959** | **41.28/0.9943** | **35.18/0.8774** |

## 4.6 ABLATION STUDY

### 4.6.1 EFFECTS OF EACH COMPONENTS

As shown in Tab. 6, we evaluate the effectiveness of various components through comparisons with the baseline method (index a). We incrementally integrate the Frequency-Modulated Module (FMM) (index b), Evolutionary Optimization Strategy (EOS) (index c), and finally, combine both components into our full EvoIR framework (index d). The average PSNR and SSIM under the three-degradation setting clearly indicate that both FMM and EOS significantly enhance the restoration performance.

Table 6: AiOIR performance of different components under the "**N+H+R**" setting.

| Index | TB | FMM | EOS | avg PSNR↑ | avg SSIM↑ |
|---|---|---|---|---|---|
| (a) | ✓ | | | 32.50 (+0.00) | 0.914 (+0.000) |
| (b) | ✓ | ✓ | | 32.68 (+0.18) | 0.916 (+0.002) |
| (c) | ✓ | | ✓ | 32.58 (+0.08) | 0.915 (+0.001) |
| (d) | ✓ | ✓ | ✓ | 33.00 (+0.50) | 0.922 (+0.008) |

### 4.6.2 EFFECTS OF EOS

To further validate the effectiveness of the proposed Evolutionary Optimization Strategy (EOS) in accelerating convergence, we provide additional analysis beyond quantitative comparisons. Specifi-

cally, we visualize both the training loss curves and the performance curves (in terms of PSNR and SSIM) over epochs, comparing scenarios with and without EOS.

As illustrated in Fig. 4, the training loss curve employing EOS exhibits a notably steeper decline in the early training stages, indicating more rapid initial convergence and efficient optimization. Correspondingly, the PSNR and SSIM curves clearly demonstrate that EOS achieves superior performance earlier and maintains consistently higher restoration quality throughout training. These results empirically confirm that EOS effectively alleviates gradient conflicts and enhances training stability, leading to faster and more robust convergence.

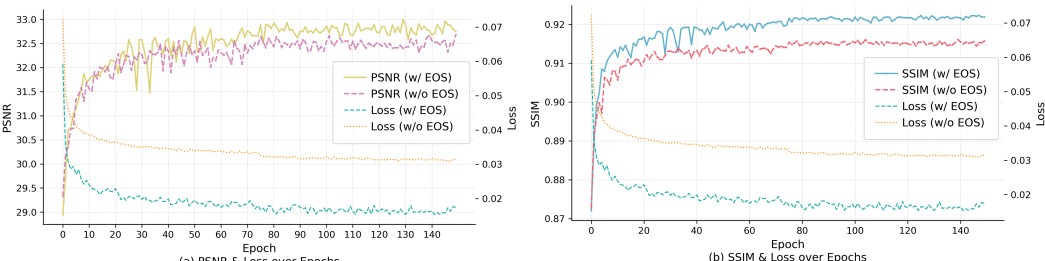

Figure 4: PSNR, SSIM and loss comparisons w/ and w/o Evolutionary Optimization Strategy (EOS) on 3D setting. We can find that model with EOS can converge faster and perform better.

To better evaluate the cost of EOS during the training stage, we instrument EOS steps and report per-epoch overhead in Tab. 7: EOS wall-clock 1.417 s (0.1% of the epoch), broken down into evaluation 1.272 s (89.8%), communication 0.094 s (6.6%), and residual 0.051 s (3.6%). EOS uses T=500, population P=5, generations G=3, and is triggered 3× per epoch (which corresponds to 24 per-rank calls across 8 GPUs). The average per global trigger is 472.3 ms, the GPU kernel time during evaluation is 390.7 ms/trigger, the payload is 94.4 MB/epoch, and the peak extra memory at EOS steps is 84.1 MB. Despite this periodic cost, Fig. 4 shows improved time-to-target PSNR/SSIM and higher final quality. In short, EOS is compute-light: on our 8×GPU setup it adds only 0.1% per-epoch wall time (1.417 s/epoch for 3 triggers) with a peak extra memory of 84.1 MB, while preserving the quality gains.

Table 7: EOS overhead per-epoch during the training stage.

| EOS wall (s) | Eval (s) | Comm (s) | Resid. (s) |
|---|---|---|---|
| 1.417 (0.1%) | 1.272 | 0.094 | 0.051 |
| Avg/call (ms) | Eval GPU (ms/call) | Payload (MB) | Peak extra mem (MB) |
| 59.0 | 48.8 | 94.4 | 84.1 |

## 5 CONCLUSION

We present EvoIR, an All-in-One Image Restoration framework unifying frequency-aware modulation and a population-based evolutionary loss scheduler. FMM performs spectral gating on low-frequency components and spatial masking on high-frequency details; EOS searches $(\alpha, \beta)$ on a held-out validation set every $T$ iterations, yielding a data-driven balance between fidelity and perception with modest overhead. To the best of our knowledge within AiOIR, EvoIR is the first work to introduce a population-based evolutionary algorithm for dynamic loss balancing. Extensive experiments demonstrate superior results across diverse degradation—achieving SOTA or highly competitive performance under multiple degradation protocols. Ablation studies attribute the gains to the synergy between FMM and EOS.

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

## A  USAGE OF LLMs

We used LLM as a general-purpose assist tool only for language polishing; all technical content is original and verified by the authors. LLMs are not authors.

## B  CODE AND MODELS

Anonymous code and models for evaluation can be found here.

## C  EXPERIMENTAL SETTINGS

### C.0.1  DATASETS

For the datasets used for AiOIR aforementioned: **N+H+R** (Noise, Haze, Rain) and **N+H+R+B+L** (Noise, Haze, Rain, Blur, Low-light). We summarize all details as follows in Tab. 8.

Table 8: Dataset summary under two training settings.

| Setting | Degradation | Training dataset (Number) | Testing dataset (Number) |
|---|---|---|---|
| **One-by-One** | Noise (**N**) | **N**: BSD400 +WED (400+4744) | **N**: CBSD68 +Urban100 +Kodak24 (68+100+24) |
| | Haze (**H**) | **H**: RESIDE-$\beta$-OTS (72135) | **H**: SOTS-Outdoor (500) |
| | Rain (**R**) | **R**: Rain100L (200) | **R**: Rain100L (100) |
| | Blur (**B**) | **B**: GoPro (2103) | **B**: GoPro (1111) |
| | Low-light (**L**) | **L**: LOL (485) | **L**: LOL (15) |
| **All-in-One** | **N+H+R** | BSD400+WED+RESIDE-$\beta$-OTS+Rain100L **Number**: 400+4744+72135+200 **Total**: 77479 | **N**: CBSD68 (68) **H**: SOTS-Outdoor (500) **R**: Rain100L (100) |
| | **N+H+R+B+L** | BSD400+WED+RESIDE-$\beta$-OTS+Rain100L +GoPro+LOL **Number**: 400+4744+72135+200+2103+485 **Total**: 80067 | **N**: CBSD68 (68) **H**: SOTS-Outdoor (500) **R**: Rain100L (100) **B**: GoPro (1111) **L**: LOL (15) |

### C.0.2  BASELINES

We compare EvoIR with a comprehensive set of baselines under both the All-in-One and One-by-One settings. We apply PSNR, SSIM as evaluation metrics. In all tables, the best and second-best results are marked in **bold** and underlined, respectively.

**All-in-One setting:** For "N+H+R" setting, we include the following recent All-in-One models: Air-Net Li et al. (2022), IDR Zhang et al. (2023), ProRes Ma et al. (2023a), PromptIR Potlapalli et al. (2023), NDR Yao et al. (2024), Gridformer Wang et al. (2024), InstructIR Conde et al. (2024), Up-Restorer Liu et al. (2025), Perceive-IR Zhang et al. (2025a), AdaIR Cui et al. (2025), MoCE-IR Zamfir et al. (2025), DFPIR Tian et al. (2025). For "N+H+R+B+L" setting, we also include TAPE Liu et al. (2022), TransWeather Valanarasu et al. (2022).

**One-by-One setting:** We adopt task-specific and general methods tailored to individual degradation types. For *Denoising*: DnCNN Zhang et al. (2017), FFDNet Zhang et al. (2018), ADFNet Shen et al. (2023), MIRNet-v2 Zamir et al. (2022a), Restormer Zamir et al. (2022b), NAFNet Chen et al. (2022); *Dehazing*: adding DehazeNet Cai et al. (2016), FDGAN Dong et al. (2020), Dehaze-Former Song et al. (2023), FSNet Cui et al. (2023a); *Deraining*: adding UMR Yasarla & Patel (2019), LPNet Gao et al. (2019), DRSformer Chen et al. (2023a); *Deblurring*: adding DeblurGAN Kupyn et al. (2018), Stripformer Tsai et al. (2022), HI-Diff Chen et al. (2023b), MPRNet Zamir et al. (2021); *Low-light enhancement*: adding URetinex Wu et al. (2022), SMG Xu et al. (2023), Retinex-former Cai et al. (2023), MIRNet Zamir et al. (2020), DiffIR Xia et al. (2023). We report All-in-One methods retrained under the One-by-One setting for comparison.

# D ADDITIONAL ABLATIONS OF BLOCK NUMBER

In this study, we investigate the impact of varying the number of FMM blocks across the four stages of EvoIR. For example, (index 6) adopts 4, 1, 1, and 1 FMM blocks in Stages 1 through 4, respectively. We report the average PSNR and SSIM across three degradation settings.

As shown in Tab. 9, (index 8) achieves the best performance in both average PSNR and SSIM. Comparisons between (index 2) & (index 3), and (index 4) & (index 5), indicate that introducing FMM blocks to multiple stages enhances restoration quality. Moreover, the parameter cost for early stages is relatively low (e.g., only 0.12M and 3.71M for Stages 1 and 2, respectively), while later stages impose significantly higher computational overhead. Notably, allocating more FMM blocks to early stages yields greater performance gains than doing so in later stages. These findings suggest that a favorable design choice for EvoIR is to prioritize block allocation in earlier stages while keeping the latter ones lightweight. The configuration in (index 8)—with 4, 2, 2, and 1 blocks—offers a balanced trade-off between performance (33.00 dB / 0.922) and model complexity (36.68M).

Table 9: Average performance of different numbers of FMM blocks in each stage.

| Index | Stage | | | | avg PSNR ↑ | avg SSIM ↑ | Params (M) |
|-------|---|---|---|---|------------|------------|------------|
|       | 1 | 2 | 3 | 4 | | | |
| (1) | 0 | 0 | 0 | 0 | 32.58 | 0.915 | 25.44 |
| (2) | 1 | 1 | 1 | 1 | 32.86 | 0.921 | 30.94 |
| (3) | 2 | 1 | 1 | 1 | 32.91 | 0.922 | 31.06 |
| (4) | 3 | 1 | 1 | 1 | 32.94 | 0.922 | 31.18 |
| (5) | 3 | 2 | 1 | 1 | 32.89 | 0.922 | 34.89 |
| (6) | 4 | 1 | 1 | 1 | 32.96 | 0.922 | 31.30 |
| (7) | 4 | 2 | 1 | 1 | 32.99 | 0.922 | 35.01 |
| (8) | 4 | 2 | 2 | 1 | 33.00 | 0.922 | 36.68 |

# E VISUALIZATION

## E.1 T-SNE VISUALIZATION FOR DEGRADATION FEATURES

We provide stage-wise visualizations that reveal how representations become degradation-aware by stages. EvoIR is a three-stage encoder-decoder architecture with a bottleneck block between encoders and decoders. After training, we take the output of Stage 1-3, perform global pooling to obtain one embedding per image, and plot t-SNE colored by degradation label for 3D AiOIR.

As is shown in Fig. 5, we anticipate that Encoder Stage 1-3 embeddings mix degradations (capturing shared low-level content), while Decoder Stage 1-3 show stronger clustering by degradation, aligning with FMM's design: spectral gating on low-frequency structure and spatial masking on high-frequency details, fused and refined deeper in the hierarchy.

## E.2 VISUAL COMPARISON UNDER THREE DEGRADATIONS

Due to page limitations, we provide additional visual comparisons for the three-degradation setting ("**N+H+R**"). As shown in Fig. 6, 7, 8, 9, and 10, we compare our EvoIR against recent state-of-the-art methods, including AdaIR Cui et al. (2025) (ICLR'25), DFPIR Tian et al. (2025) (CVPR'25), and MoCE-IR Zamfir et al. (2025) (CVPR'25).

Across all degradation types, EvoIR consistently preserves higher fidelity while maintaining fine textures and structural details. The zoomed-in regions further highlight EvoIR's superiority in recovering sharp edges and realistic patterns compared to other approaches. These visual results are consistent with the quantitative improvements reported earlier.

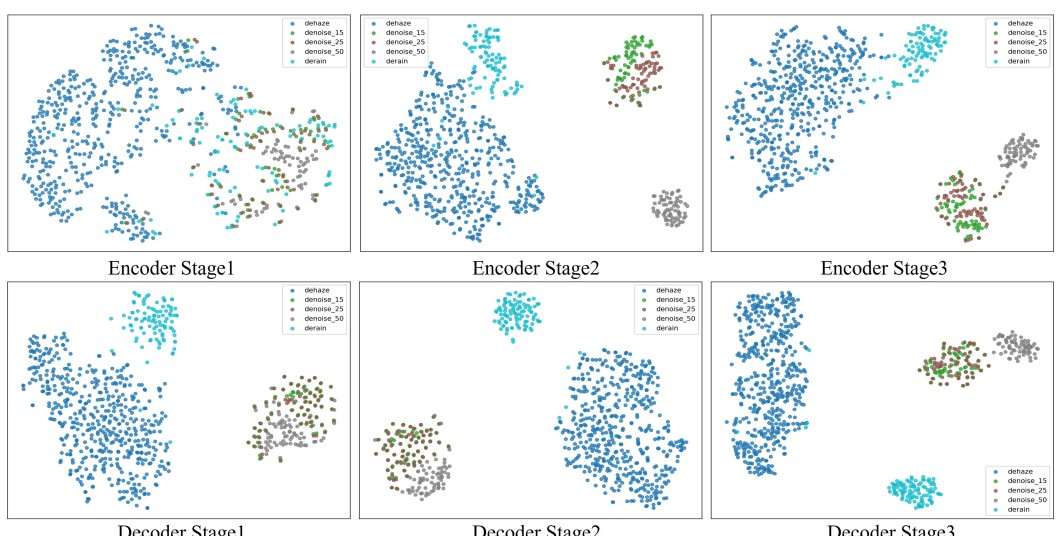

Figure 5: The t-SNE visualization of features after each stage of encoders and decoders.

### E.3 VISUAL COMPARISON UNDER FIVE DEGRADATIONS

As some methods do not provide visual results under the five-degradation setting ("**N+H+R+B+L**"), we select InstructIR Conde et al. (2024) (ECCV'24) and MoCE-IR Zamfir et al. (2025) (CVPR'25) for comparison with our EvoIR.

Zoom-in views in Fig. 11, 12, 13, 14, and 15 reveal that EvoIR better preserves both textural and structural details, producing results that are visually closer to the reference images. These observations indicate better restoration quality compared to the competing methods.

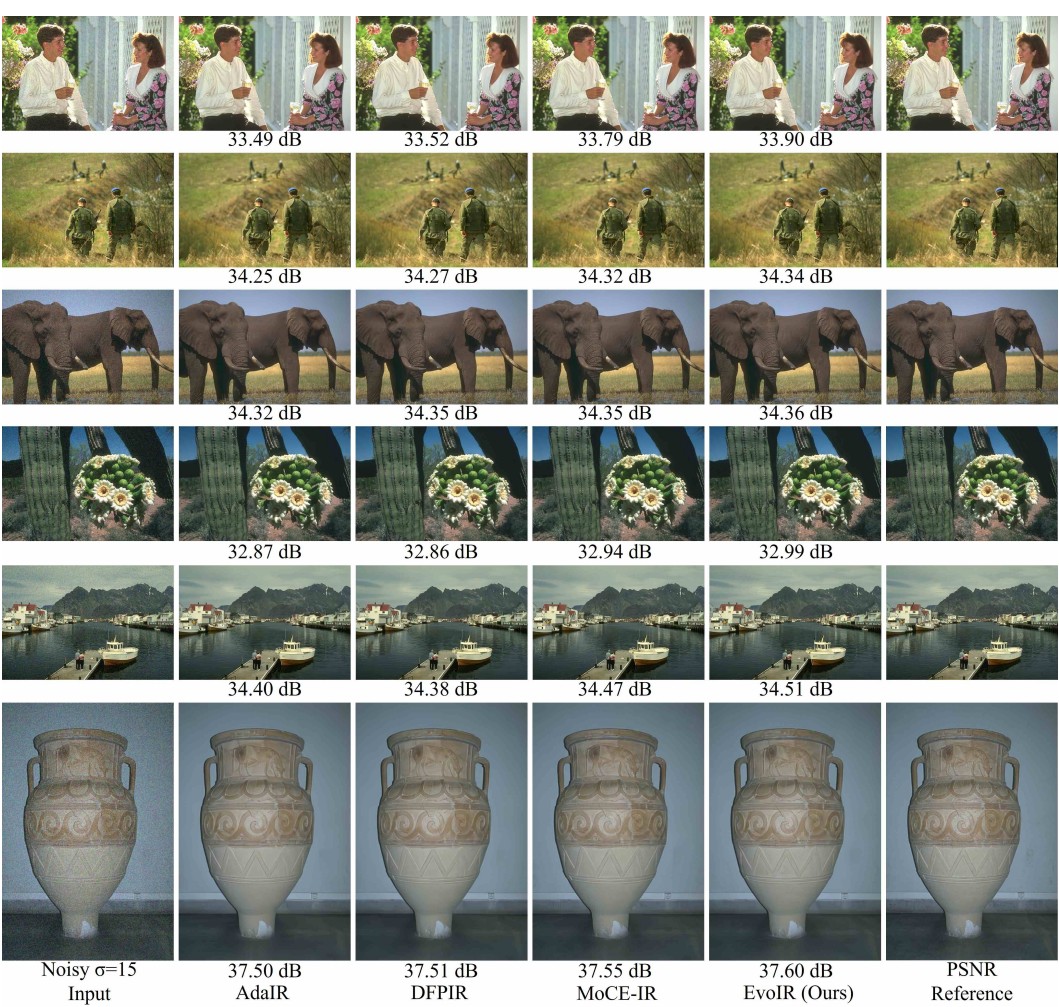

Figure 6: Denoising ($\sigma = 15$) visual comparisons of EvoIR with state-of-the-art All-in-One methods under "**N+H+R**" setting.

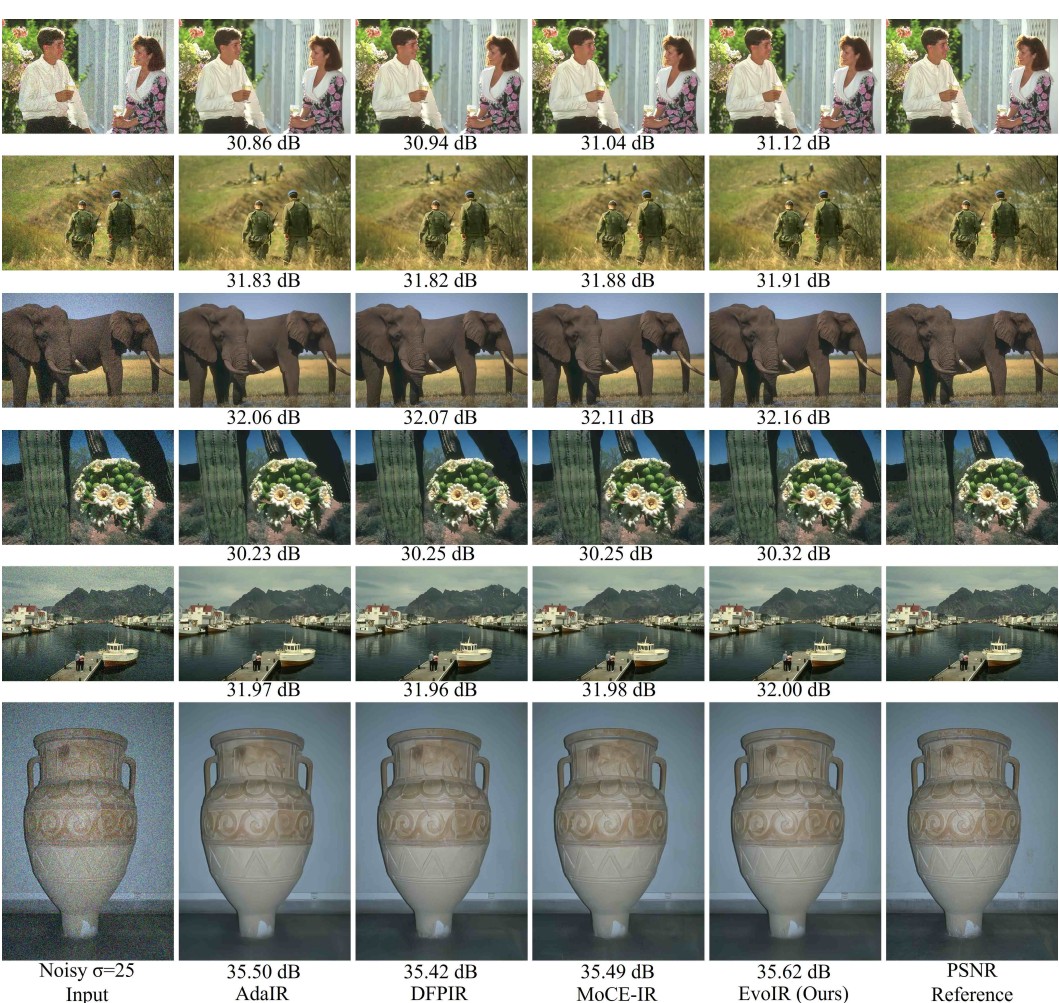

Figure 7: Denoising ($\sigma = 25$) visual comparisons of EvoIR with state-of-the-art All-in-One methods under "**N+H+R**" setting.

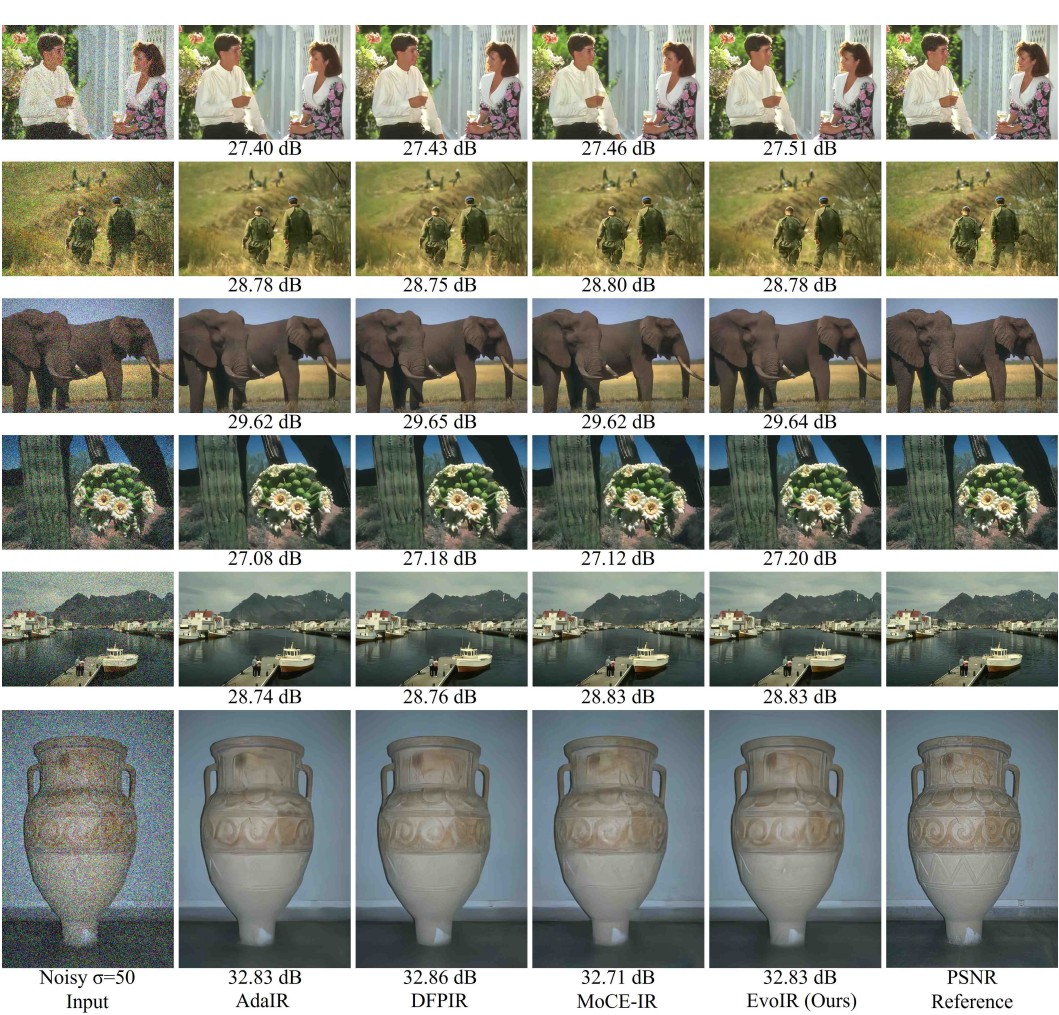

Figure 8: Denoising ($\sigma = 50$) visual comparisons of EvoIR with state-of-the-art All-in-One methods under "**N+H+R**" setting.

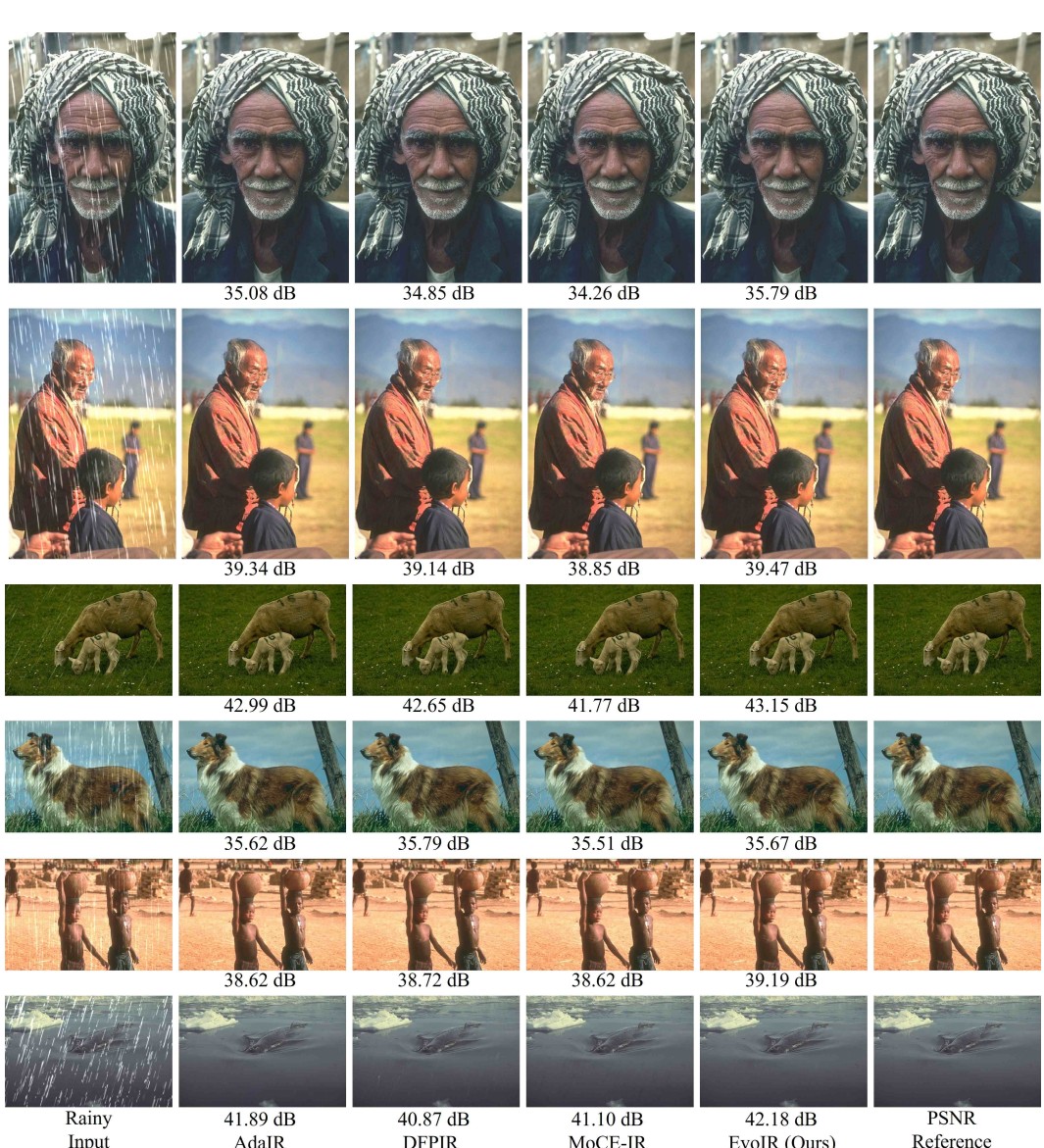

|  | 35.08 dB | 34.85 dB | 34.26 dB | 35.79 dB |  |
|  | 39.34 dB | 39.14 dB | 38.85 dB | 39.47 dB |  |
|  | 42.99 dB | 42.65 dB | 41.77 dB | 43.15 dB |  |
|  | 35.62 dB | 35.79 dB | 35.51 dB | 35.67 dB |  |
|  | 38.62 dB | 38.72 dB | 38.62 dB | 39.19 dB |  |
| Rainy Input | 41.89 dB AdaIR | 40.87 dB DFPIR | 41.10 dB MoCE-IR | 42.18 dB EvoIR (Ours) | PSNR Reference |

Figure 9: Deraining visual comparisons of EvoIR with state-of-the-art All-in-One methods under "**N+H+R**" setting.

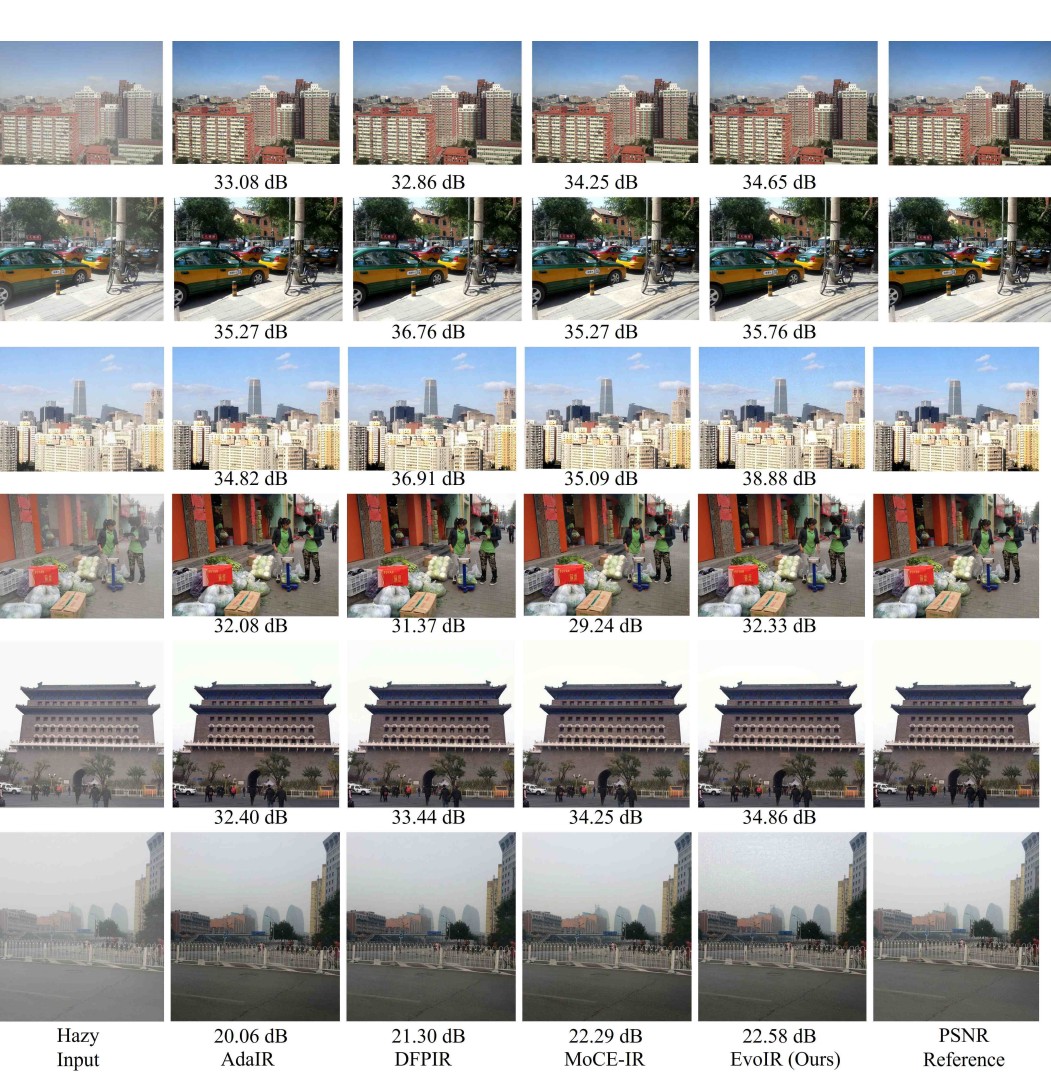

Figure 10: Dehazing visual comparisons of EvoIR with state-of-the-art All-in-One methods under "**N+H+R**" setting.

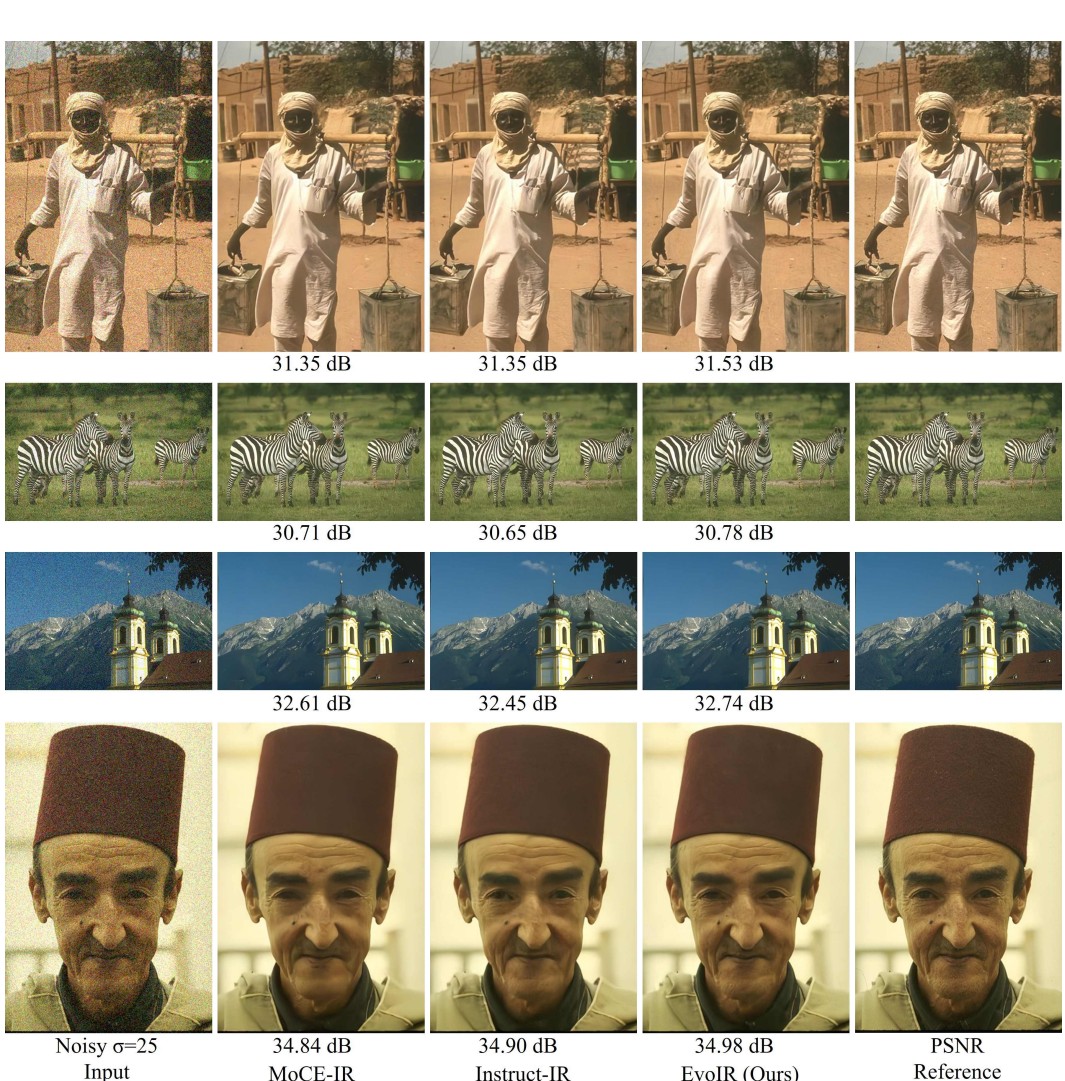

Figure 11: Denoising ($\sigma = 25$) visual comparisons of EvoIR with state-of-the-art All-in-One methods under "**N+H+R+B+L**" setting.

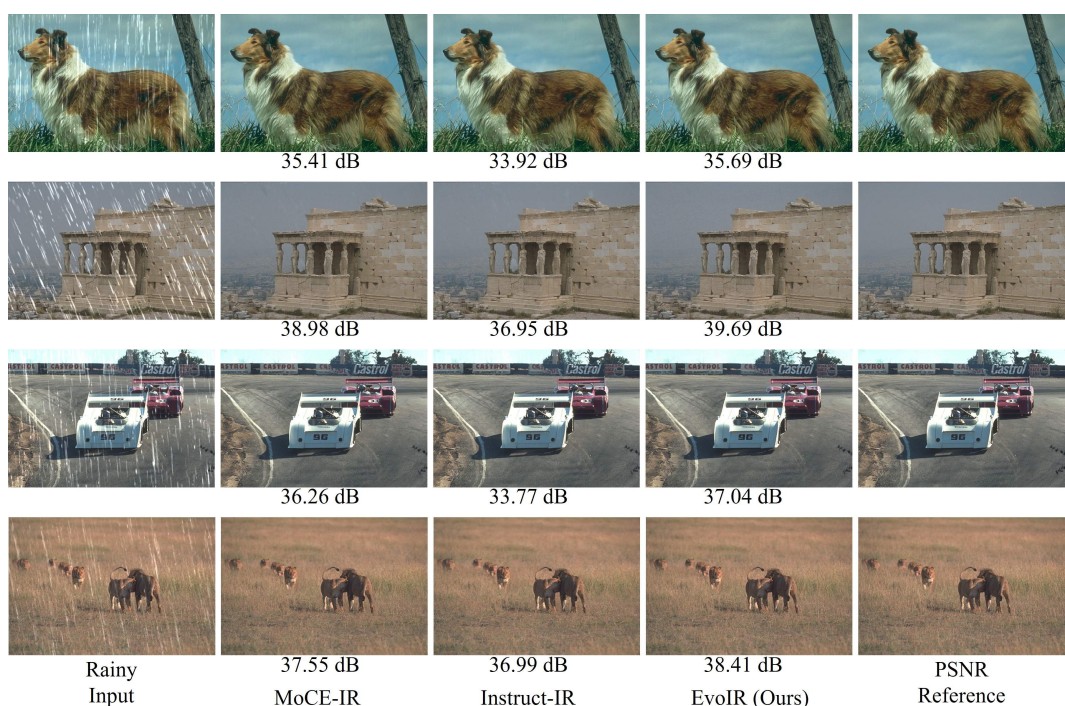

Figure 12: Deraining visual comparisons of EvoIR with state-of-the-art All-in-One methods under "**N+H+R+B+L**" setting.

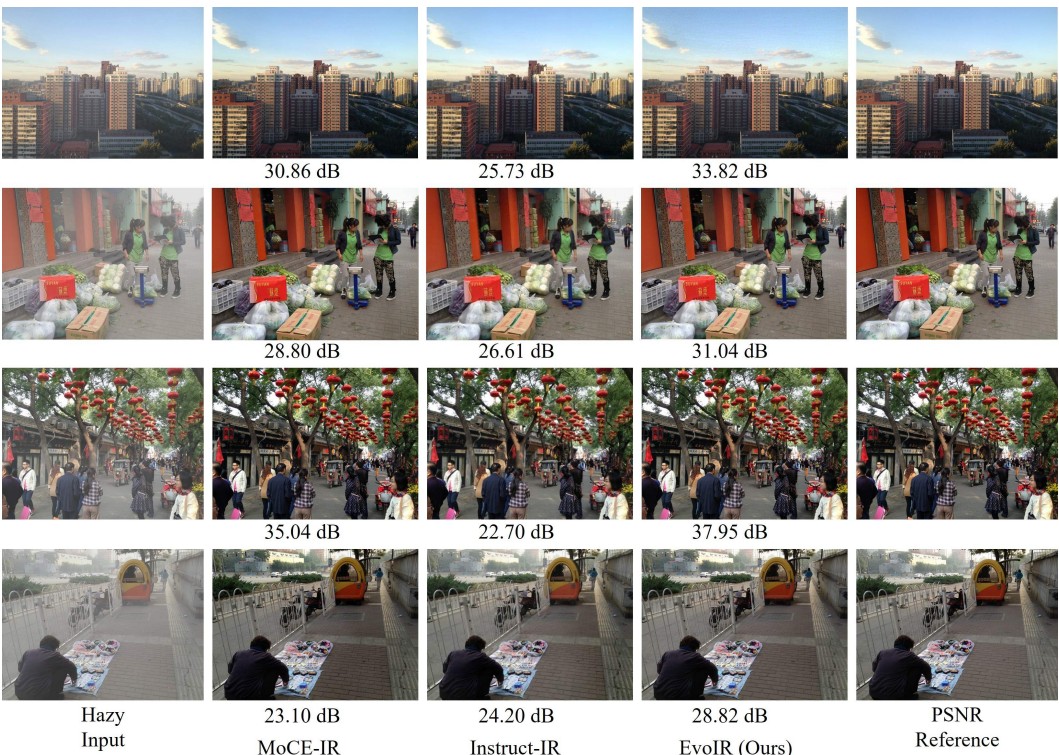

Figure 13: Dehazing visual comparisons of EvoIR with state-of-the-art All-in-One methods under "**N+H+R+B+L**" setting.

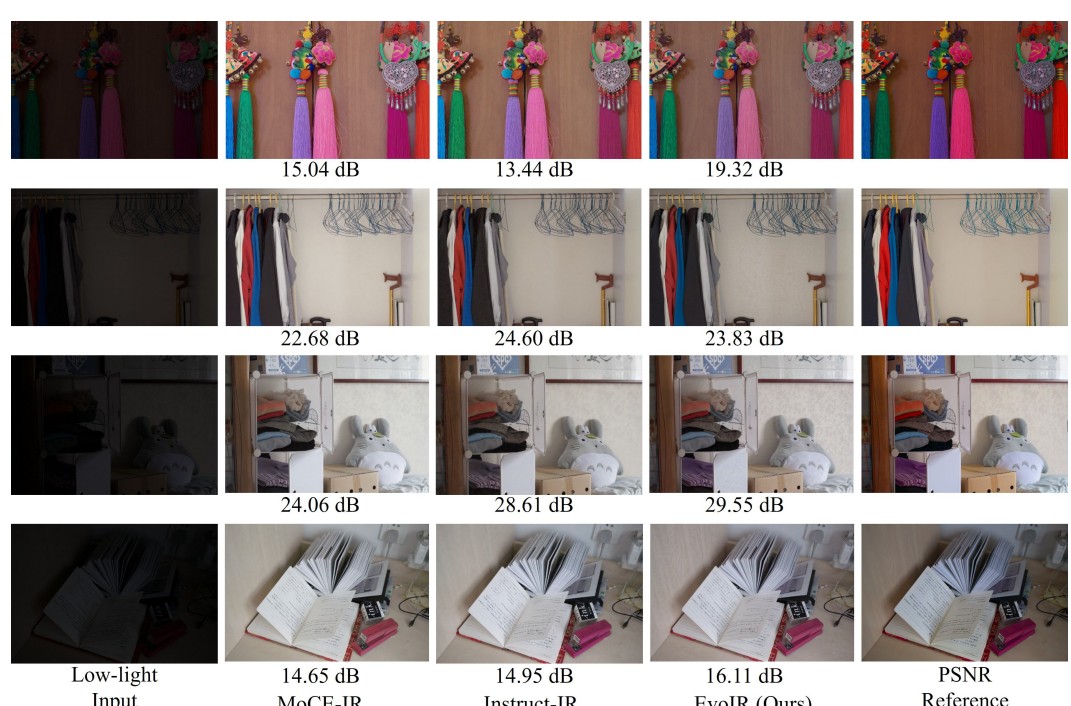

Figure 14: Enhancement visual comparisons of EvoIR with state-of-the-art All-in-One methods under "**N+H+R+B+L**" setting.

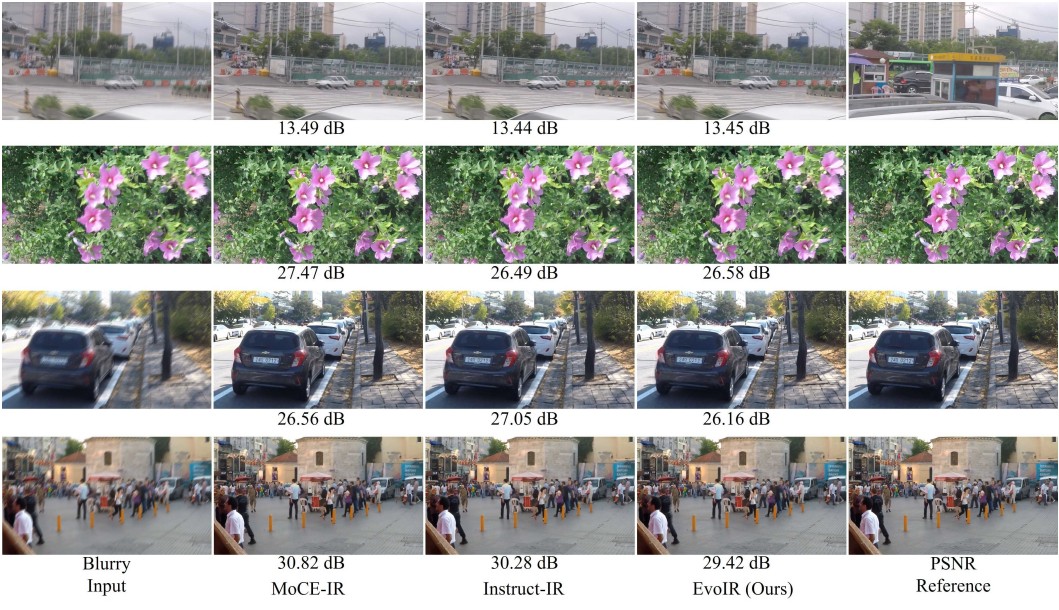

Figure 15: Deblurring visual comparisons of EvoIR with state-of-the-art All-in-One methods under "**N+H+R+B+L**" setting.