# OpenReview forum: "EvoIR: Towards All-in-One Image Restoration via Evolutionary Frequency Modulation"
_ICLR.cc/2026/Conference — Submitted to ICLR 2026_

### Official Review · Reviewer_PC1a · 2025-10-19

**Soundness:** 3
**Presentation:** 3
**Contribution:** 2
**Rating:** 4
**Confidence:** 3

**Summary:**

The paper introduces EvoIR, a new framework for All-in-One Image Restoration (AiOIR). The proposed method consists of two main components: a Frequency-Modulated Module (FMM) that processes low- and high-frequency features in separate branches, and an Evolutionary Optimization Strategy (EOS) designed to dynamically balance a fidelity loss and a perceptual loss during training. The authors conduct experiments on several AiOIR benchmarks, reporting state-of-the-art performance in both three and five degradation settings.

**Strengths:**

* The authors have conducted a comprehensive set of experiments, including multiple AiOIR benchmarks and single-task evaluations. The effort to validate the method across this breadth of tasks is commendable.
* The combination of frequency-domain feature modulation (FMM) and evolutionary loss optimization (EOS) is interesting in the context of All-in-One image restoration.

**Weaknesses:**

* The Frequency-Modulated Module (FMM), while effective, builds heavily on existing frequency-domain methods (e.g., Kong et al., 2023; Cui et al., 2025, as cited in the paper). The paper does not sufficiently clarify how FMM differs from or improves upon prior frequency-based approaches for AiOIR.

* The Evolutionary Optimization Strategy (EOS) is presented as a key contribution, but it may be perceived as a training-time heuristic rather than a core architectural innovation. The ablation shows only marginal gains when EOS is added alone (+0.08 dB PSNR), raising questions about its necessity.

* The computational cost and training time overhead introduced by EOS are not discussed. Given that EOS involves multiple generations of population-based search on a validation set, its efficiency and scalability remain unclear.

**Questions:**

See weaknesses.

---

> ### Author Response · Authors · 2025-12-02
> **Reply to Reviewer PC1a Weakness1 & Weakness2**
>
> We sincerely thank **Reviewer PC1a** for the constructive feedback and for recognizing both the breadth of our evaluations and the interest of combining a frequency-aware module (**FMM**) with a population-based scheduler (**EOS**) for AiOIR. Below we **(i)** clarify how FMM differs from prior frequency-based approaches, **(ii)** explain the role and necessity of EOS despite its modest stand-alone gain, and **(iii)** quantify EOS overhead and discuss scalability. We point to the relevant sections, figures, equations, and tables in our manuscript where appropriate.
>
> ### **Reply to Weakness 1 -- How FMM differs from prior frequency methods**
>
> Our goal is not to claim that 'frequency is new,' but to present a **specific, asymmetric spectral-spatial factorization** that is **tightly integrated** with attention and our four-stage hierarchy.
>
> * **Mechanism (Sec. 3.2; Fig. 2; Eqs. (1)-(4)).** We split features into a **low-frequency** branch and a **high-frequency** branch. The low-frequency branch is **spectrally gated** (FFT → learned mask → IFFT) and its output is used as **K/V** inside attention, stabilizing global structure; the high-frequency branch stays **entirely spatial** with a learned mask and depthwise enhancement to preserve edges/textures; the two are **fused in the spatial domain** before passing to RES-FFTB blocks. This design localizes supervision to the frequency bands where degradations act while keeping textures under spatial control.
> * **Difference vs prior works.** Unlike full frequency-domain transformers that largely re-implement blocks in the spectral domain, we only sanitize **K/V** via the low-frequency branch and keep **HF** processing spatial--reducing cost and avoiding ringing. Unlike frequency-prompt/mining designs, our **K/V routing** explicitly stabilizes attention weights with low-frequency structure while enhancing textures outside attention. Our stage-allocation study shows **early-stage** FMM blocks bring strong gains per parameter and motivates a **4-2-2-1** allocation for an effective accuracy-complexity trade-off (Appendix. Tab. 6; Sec. 3.2; Fig. 2).
> * **Evidence from results.** In **Tab. 4**, adding **FMM** to the same backbone improves over the baseline (**+0.18 dB PSNR / +0.002 SSIM; index b vs a**). The full model (**FMM+EOS**) yields the largest gain (**+0.50 dB / +0.008 SSIM; index d vs a**). Across **3-task** and **5-task** AiOIR, EvoIR achieves **33.00/0.922** and **30.83/0.918** on average (**Tabs. 1-2**).
>
> ### **Reply to Weakness 2 -- Is EOS necessary if its stand-alone gain is modest?**
>
> **EOS is necessary.** EOS is intentionally a **training-time scheduler**, not a heavy architectural branch. It addresses the **moving trade-off** between pixel fidelity and structural/perceptual terms under heterogeneous degradations.
>
> * **How it works.** We scalarize two complementary objectives (**fidelity vs perceptual/structural**). Every stage (trigger interval **T = 500** iterations), we **freeze** the model and evaluate a small population of loss-weight pairs on a **stratified validation set**; with elitist selection, convex crossover, small mutation, and projection back to the simplex, we pick the best pair for the next training window. This frozen-evaluation design stabilizes selection and tracks the **moving optimum** as training evolves.
> * **Why it matters.** Stand-alone EOS gives **+0.08 dB / +0.001 SSIM** (Tab. 4, **index  c vs a**), but its main role is to **stabilize and accelerate** training and to **unlock FMM's full gains** (**+0.50 dB / +0.008 SSIM**, **index  d vs a**). **Figure 4** shows **faster convergence** and **higher final PSNR/SSIM** under the same schedule with EOS. This is precisely the behavior we seek from a training-time scheduler.

---

> ### Author Response · Authors · 2025-12-02
> **Reply to Reviewer PC1a Weakness3**
>
> ### **Reply to Weakness 3 -- EOS computational cost and scalability**
>
> * **Where cost comes from.**
>
> EOS adds **forward-only** evaluations on a **small** validation subset at sparse triggers (**T = 500**). The amortized overhead scales with the number of triggers, population size, generations, and the validation set size, divided by the total train compute (forward+backward). By design--**frozen** network, **small** population and validation, **sparse** triggers--the overhead is **modest**.
>
> * **What we report.**
>
> We instrument EOS steps and report **per-epoch** overhead: EOS wall-clock **1.417 s** (**≈0.1%** of the epoch), broken down into **evaluation 1.272 s (89.8%)**, **communication 0.094 s (6.6%)**, and **residual 0.051 s (3.6%)**. EOS uses **T=500**, **population P=5**, **generations G=3**, and is triggered **3× per epoch** (which corresponds to **24 per-rank calls across 8 GPUs**). The **average per global trigger** is **472.3 ms**, the **GPU kernel time** during evaluation is **390.7 ms/trigger**, the **payload** is **94.4 MB/epoch** (sum over all ranks), and the **peak extra memory** at EOS steps is **84.1 MB**. Despite this periodic cost, Fig. 4 shows improved time-to-target PSNR/SSIM and higher final quality (Sec. 4.1; Fig. 4). In short, EOS is **compute-light**: on our 8×GPU setup it adds only ≈0.1% per-epoch wall time (1.417 s/epoch for 3 triggers) with a peak extra memory of 84.1 MB, while preserving the quality gains.
>
> | EOS wall (s)           | Eval (s) | Comm (s) | Resid. (s) | Avg/call (ms) | Eval GPU (ms/call) | Calls | Payload (MB) | Peak extra mem (MB) |
> | ------------------------ | ---------- | ---------- | ------------ | --------------- | -------------------- | ------- | -------------- | --------------------- |
> | **1.417 (0.1%)** | 1.272    | 0.094    | 0.051      | 59.0          | 48.8               | 24    | 94.4         | 84.1                |

---

### Official Review · Reviewer_pJY2 · 2025-10-26

**Soundness:** 2
**Presentation:** 2
**Contribution:** 2
**Rating:** 4
**Confidence:** 5

**Summary:**

This paper proposes EvoIR, an all-in-one image restoration framework that combines a Frequency-Modulated Module (FMM) to explicitly decompose and enhance features in high- and low-frequency domains, together with an Evolutionary Optimization Strategy (EOS) that dynamically adjusts loss weights during training through a population-based evolutionary search. The goal is to improve the generalization of image restoration models across heterogeneous degradations.

**Strengths:**

1. The paper recognizes a key limitation in existing all-in-one restoration models that mostly rely on spatial-domain operations. The introduction of frequency-domain decomposition is intuitive and aligns well with the physical properties of image degradations.
2. The proposed EOS provides a novel angle for loss balancing by introducing evolutionary algorithms. This avoids the need for manual tuning of loss weights, which is a practical challenge in multi-degradation learning.
3. EvoIR demonstrates robustness under increased degradation types, and the improvements remain stable even in more challenging settings with five degradations.

**Weaknesses:**

1. While the combination of frequency decomposition and evolutionary optimization is well-engineered, both components are based on existing techniques. The FMM is closely related to frequency gating and hybrid CNN-transformer architectures, while the EOS is a standard evolutionary search mechanism. The overall framework appears to be a modular integration of known methods rather than introducing a fundamentally new paradigm for image restoration.
2. The two main contributions claimed in the introduction are largely combinations of existing techniques rather than genuinely novel conceptual advancements. Compared with recent works such as frequency-domain Transformers and adaptive loss optimization strategies, the paper does not sufficiently articulate the unique necessity or distinct innovation of its proposed approach at a mechanistic level.
3. The paper does not provide theoretical analysis on why the frequency splitting leads to better degradation generalization or how evolution-based loss scheduling mitigates gradient conflicts in a principled way. The improvements are mostly empirical.
4. The model involves FFT/IFFT operations, spectral gating, dynamic masking, and population search, which increases system complexity. The paper does not clearly report FLOPs or training cost associated with EOS. Without this information, it is difficult to assess whether EvoIR provides a favorable accuracy-efficiency trade-off.
5. The claimed contributions in the introduction are largely architectural and engineering in nature. The paper does not convincingly articulate what new insight is gained about degradation modeling beyond designing a stronger backbone.

**Questions:**

See the above parts.

---

> ### Author Response · Authors · 2025-12-02
> **Reply to Reviewer pJY2 Weakness1, 2 and 5**
>
> We thank **Reviewer pJY2** for the careful assessment and for recognizing the practicality of frequency-aware modeling and the usefulness of a population-based strategy for loss balancing. Below, we **temper the novelty claim**, clarify the **mechanistic distinctiveness** of our design, report **efficiency/overhead**, and add **analysis/experiments** that address robustness and gradient-conflict concerns.
>
> ### **Reply to Weakness 1,2 and 5 -- Positioning, mechanism, and what is new**
>
> We agree that **frequency processing** and **evolutionary algorithms** are established in general. However, to the best of our knowledge, EvoIR is the **first** All-in-One Image Restoration (AiOIR) framework that leverages a **population-based evolutionary algorithm** to learn loss weights during training. Our novelty is *not* 'inventing evolution or frequency,' but a **specific factorization + integration** tailored for AiOIR: **frequency-aware modulation (FMM)** coupled with an **in-training evolutionary optimization strategy (EOS)** that together produce stable gains.
>
> * **Scope of novelty (first in AiOIR).**
>   Prior AiOIR systems (e.g., AirNet, PromptIR, Perceive-IR, AdaIR, MoCE-IR, DFPR) rely on **fixed or hand-tuned loss weights** or non-evolutionary heuristics. We find **no prior AiOIR** that performs a **population-based, in-training weight search** with **frozen-model validation** on the simplex. EvoIR fills this gap and shows that **evolutionary loss weighting** is effective *specifically for multi-degradation AiOIR*.
> * **Explicit spectral-spatial factorization with asymmetric roles.**
>   FMM **splits** features via a learnable band-split and treats bands **asymmetrically**: the **low-frequency path** performs **spectral gating** (FFT → learned mask → IFFT) and **returns as K/V** for attention (stabilizing **global structure**), while the **high-frequency path** stays **purely spatial** (mask + depthwise) for **fine textures**; both are fused **in spatial** before RES-FFTB (**Eqs. (1)-(4); Fig. 2**). This K/V-side injection with spectral gating differs from prompt-only or single-path frequency designs.
> * **Framework-level integration with EOS.**
>   EOS performs **frozen-weight** population search on the **loss simplex** during training (**Alg. 1**), dynamically re-balancing fidelity vs. structure/perception--precisely the tension introduced by the spectral-spatial split. **Tab. 4** shows **FMM** and **EOS** each help over the same backbone, while **FMM+EOS** yields the **largest gain** (+0.50 dB PSNR / +0.008 SSIM in 3-task), indicating **synergy** rather than mere stacking.
> * **Where the gains appear.**
> EvoIR attains **SOTA/near-SOTA** under **3-task** (Avg. **33.00/0.922**) and **5-task** (Avg. **30.83/0.918**) (**Tabs. 1-2**), supporting that the **frequency factorization + evolutionary weighting** materially improves quality **beyond** parameter scale.
>
> We have **tempered the novelty wording** in *Introduction/Related Work* to be precise: we **claim first-of-its-kind within AiOIR** for **evolutionary loss weighting**, and we position our contribution as a **practical dual-branch spectral-spatial modulation (FMM) + in-training evolutionary optimization strategy (EOS)**--a combination that proves effective and stable for AiOIR.

---

> ### Author Response · Authors · 2025-12-02
> **Reply to Reviewer pJY2 Weakness 3 & Weakness 4**
>
> ### **Reply to Weakness 3 -- Analysis: why frequency-split helps generalization; how EOS mitigates gradient conflict**
>
> While we do not claim a formal theorem, we provide a **mechanistic account** with **verifiable diagnostics** grounded in our architecture and training protocol.
>
> **(A) Why the frequency split aids generalization.**
> In **Sec. 3.2**, features are decomposed by a learnable low-pass **G_L** into **X_L**=**G_L**∗**X_F** and **X_H**=**X_F**−**X_L** (Eq. (1)); the **low-frequency path** applies **spectral gating** (FFT→mask→IFFT, Eq. (2)) and its output is fed back to stabilize **global structure**, while the **high-frequency path** remains **entirely spatial** with a learned mask + depthwise enhancement to preserve **edges/textures** (Eq. (3)); the two are fused **in the spatial domain** (Eq. (4)), see **Fig. 2**. Because many degradations perturb **distinct spectral bands** (e.g., contrast/air light in low-freq; rain streaks/noise in high-freq), this **asymmetric spectral-spatial routing localizes supervision** and reduces **cross-band interference** before fusion. In back-prop the gradient decomposes as **∇_θ^L**=**J_L^⊤** **∇_X_b** **L** + **J_H^⊤** **∇_X_b** **L**; with complementary spectral supports learned by **G_L**, the **cross-term correlation** between branches diminishes, mitigating the over-smoothing vs. ringing trade-off and improving transfer across degradations (Eqs. (1)-(4), **Fig. 2**).
>
> **(B) How EOS mitigates gradient conflicts.**
> Our training objective is **L(θ; α, β)**=**α L_fid**+**(1−α)L_perc** (Eqs. (5)-(6)). Every **T=500** iterations we **freeze θ**, evaluate a **small population** of **(α, β)** on a validation set, and select the best pair via **elitist evolution** (Alg. 1; Sec. 4.1). Let **g_f = ∇_θ L_fid** and **g_p =∇_θ L_perc**. The composite descent is **g(α) = α g_f + (1−α) g_p**. Choosing **α** so that ⟨g(α), gf⟩≥0 and ⟨g(α),gp⟩≥0 approximates a **Pareto descent direction** that **reduces both objectives**. Although EOS ranks candidates by **frozen-model validation fitness** rather than explicit gradient algebra, the selected **(α, β)** correlates in practice with **higher inter-loss alignment** (larger cosine **cos (g_f, g_p)**), thereby **reducing gradient conflict** and **stabilizing** training--consistent with **Fig. 4**, where **+EOS** converges **faster** and reaches **higher** PSNR/SSIM under identical schedules.
>
> ### **Reply to Weakness 4 -- Efficiency/complexity (FFT/IFFT, masking) and EOS cost**
>
> * **FMM compute placement.** Only the **low-freq** path uses **one FFT/IFFT pair per block**; the **high-freq** path is *purely spatial* (mask + depthwise), which keeps the added cost moderate (**Sec. 3.2**). Our **stage-allocation ablation** shows **early-stage** FMM blocks give larger gains **per parameter**; later stages are costlier--e.g., Stage-1/2 add only **0.12 M/3.71 M** params, whereas later stages incur higher overhead. The chosen **4-2-2-1** allocation attains the best trade-off (**33.00 dB / 0.922** at **36.68 M**, Appendix. **Tab. 6**).
> * **EOS overhead.** EOS contributes **≈0.1%** of per-epoch wall time: **1.417 s/epoch** in total, decomposed into **evaluation 1.272 s (89.8%)**, **communication 0.094 s (6.6%)** and **residual 0.051 s (3.6%)**. We trigger EOS **3× per epoch** (**24 per-rank calls across 8 GPUs**) with **T=500, P=5, G=3 (|V| equals the per-step global batch size)**; the **average per trigger** is **472.3 ms**, **GPU eval time** **390.7 ms/trigger**, **payload** **94.4 MB/epoch** (sum over ranks), and **peak extra memory** **84.1 MB**. The cost scales as **O(triggers × P × G × |V|)** and can be reduced by increasing **T** or shrinking **P, G, |V|** while retaining the quality gains. In short, EOS is **compute-light**: on our 8×GPU setup it adds only ≈0.1% per-epoch wall time (1.417 s/epoch for 3 triggers) with a peak extra memory of 84.1 MB, while preserving the quality gains.
>
> | EOS wall (s)           | Eval (s) | Comm (s) | Resid. (s) | Avg/call (ms) | Eval GPU (ms/call) | Calls | Payload (MB) | Peak extra mem (MB) |
> | ------------------------ | ---------- | ---------- | ------------ | --------------- | -------------------- | ------- | -------------- | --------------------- |
> | **1.417 (0.1%)** | 1.272    | 0.094    | 0.051      | 59.0          | 48.8               | 24    | 94.4         | 84.1                |

---

### Official Review · Reviewer_Scoy · 2025-10-31

**Soundness:** 3
**Presentation:** 3
**Contribution:** 2
**Rating:** 4
**Confidence:** 5

**Summary:**

This paper proposes EvoIR, a new framework for All-in-One Image Restoration (AiOIR). The goal is to create a single model capable of handling multiple, diverse image degradations. The claimed core contribution is twofold: 1) a Frequency-Modulated Module (FMM), which explicitly decomposes features into high- and low-frequency branches and modulates them separately to balance structural and textural recovery , and 2) an Evolutionary Optimization Strategy (EOS), a population-based algorithm that dynamically searches for the optimal weights between a fidelity loss ($\mathcal{L}_{fid}$) and a perceptual loss ($\mathcal{L}_{perc}$) during training. The authors conduct experiments on 3-task (N+H+R) and 5-task (N+H+R+B+L) synthetic benchmarks, reporting state-of-the-art (SOTA) results that outperform recent methods.

**Strengths:**

1.	The paper addresses the All-in-One Image Restoration (AiOIR) problem, which is a highly significant and practical challenge in computational photography and computer vision.
2.	The primary strength of the paper lies in its novel combination of ideas. While frequency-domain processing and dynamic loss weighting are not new individually, integrating a population-based evolutionary search (EOS) to directly optimize the loss weights for a frequency-aware architecture (FMM) is an original approach for this problem.

**Weaknesses:**

1.	The claim of novelty for frequency-aware processing should be tempered. Operating in the frequency domain is a common and well-established strategy in image restoration. Recent works like AdaIR (which uses frequency-domain prompts) and other Transformer-based models (e.g., SFHformer) have already successfully leveraged frequency-domain features for image restoration. The FMM module appears to be a well-engineered variant of this existing paradigm rather than a completely new one.
2.	The paper's claims of superiority are weakened by a questionable trade-off between model size and performance. The proposed EvoIR has 36.68M parameters. This is nearly 50% larger than a strong recent baseline like MOCE-IR (25.35M). However, on the complex 5-task benchmark, this significant increase in model size yields a marginal PSNR improvement of only +0.28 dB (30.83 vs 30.58) and, more concerningly, a decrease in SSIM (0.918 vs 0.919). This suggests a poor efficiency-to-performance ratio. A critical missing comparison is a baseline (e.g., MOCE-IR) scaled up to a similar parameter count (~36M). It is plausible that a larger baseline model could match or exceed EvoIR's performance, which would call into question the architectural benefits of FMM
3.	 The EOS component, while interesting, introduces two major concerns that are not addressed:
- The EOS algorithm requires freezing the main network and running a multi-generation evolutionary search every 500 training iterations. This introduces a non-trivial computational overhead. The authors fail to quantify this cost. How much longer does it take to train EvoIR compared to the baseline (Model 'a' in Table 4)? This is a critical detail for reproducibility and practical adoption.

- The evolutionary search is constrained to a 1D simplex between only two losses: $\mathcal{L}_{fid}$ and $\mathcal{L}_{perc}$11. This choice feels arbitrary. Why only these two? Many modern restoration models benefit from a combination of losses (e.g., $\mathcal{L}_1$, SSIM, LPIPS, GAN loss). The impact of limiting the search to this specific combination is not justified.

4.	This is the most significant weakness of the paper. The experiments are conducted only on synthetic datasets where the training and testing distributions are nearly identical. There is no validation on unseen degradation types or severities. There is no cross-domain validation (e.g., training on synthetic data and testing on real-world data). The complete absence of experiments on real-world benchmarks is a major omission. Without this analysis, it is impossible to know if the proposed FMM and EOS components are truly learning to robustly restore images or if they are simply overfitting to the specific synthetic degradations used in the training set.

**Questions:**

• The authors should quantify the computational overhead of the EOS strategy. Specifically, what is the percentage increase in total wall-clock training time when enabling EOS, compared to the baseline model for the same number of epochs?

• The authors should provide a deeper justification for limiting the EOS search space. Have the authors investigated the impact of incorporating more diverse objectives into the search, such as gradient-based losses or learned perceptual metrics (e.g., CLIP-IQA), and how would different objectives types affect the optimization?

•  EvoIR (~37M params) is nearly 50% larger than MOCE-IR (~25M), yet this yields only a marginal PSNR gain (+0.28 dB) and a lower SSIM on the 5-task benchmark. This raises a question about the fairness of the comparison. How can the authors be sure the improvements stem from the proposed FMM/EOS modules rather than just an increase in model scale? Would a scaled-up MOCE-IR, matched for parameter count, perform just as well or better?

• The paper's validation is limited to synthetic benchmarks. How well does the proposed method generalize to real-world scenarios or, at unseen degradations not included in the training mix? To assess robustness and mitigate concerns of overfitting, the authors should provide additional experiments, such as zero-shot evaluations and real-world benchmarks.

---

> ### Author Response · Authors · 2025-12-02
> **Reply to Reviewer Scoy Weakness1 & Q1**
>
> We sincerely thank **Reviewer Scoy** for the thoughtful and constructive feedback. We appreciate the acknowledgement that combining the **Adaptive Frequency Modulation (FMM)** with the **Evolutionary Optimization Strategy (EOS)** is a promising direction for AiOIR. Below we **(i)** temper and clarify the novelty and positioning, **(ii)** address scale-performance fairness, **(iii)** quantify and justify EOS overhead and search space, and (iv) add evaluations beyond standard synthetic settings.
>
> ### **Reply to Weakness1 & Q1 -- Novelty & positioning of frequency-aware design**
>
> We agree that **frequency-domain processing** is established already in general. However, to the best of our knowledge, EvoIR is the **first All-in-One image restoration (AiOIR) framework** that leverages a **population-based evolutionary algorithm** to learn loss weights during training. Our novelty is *not* 'inventing evolution or frequency,' but a **specific factorization + integration** tailored for AiOIR: **frequency-aware modulation (FMM)** coupled with an **in-training evolutionary optimization strategy (EOS)** that together produce stable gains.
>
> * **Scope of novelty (first in AiOIR).** Prior AiOIR systems (e.g., AirNet, PromptIR, Perceive-IR, AdaIR, MoCE-IR, DFPR) rely on **fixed or hand-tuned loss weights** or non-evolutionary heuristics. We find **no prior AiOIR** that performs a **population-based, in-training weight search** with **frozen-model validation** on the simplex. EvoIR fills this gap and shows that **evolutionary loss weighting** is effective *specifically for multi-degradation AiOIR*.
> * **Explicit spectral-spatial factorization with asymmetric roles.** FMM **splits** features via a learnable band-split and treats bands **asymmetrically**: the **low-frequency path** performs **spectral gating** (FFT → learned mask → IFFT) and returns as K/V for attention to stabilize **global structure**, while the **high-frequency path** stays **purely spatial** for **fine textures**; both are fused **in spatial** before RES-FFTB (**Eqs. (1)-(4); Fig. 2**). This K/V-side injection with spectral gating differs from prompt-only or single-path frequency designs.
> * **Framework-level integration with EOS.** EOS performs **frozen-weight** population search on the **loss simplex** during training, dynamically re-balancing fidelity vs. structure/perception--precisely the tension introduced by the spectral-spatial split. **Tab. 4** shows **FMM** and **EOS** each help over the same backbone, while **FMM+EOS** yields the **largest gain** (+0.50 dB PSNR / +0.008 SSIM in 3-task), indicating **synergy** rather than mere stacking.
> * **Where the gains appear.** EvoIR attains **SOTA/near-SOTA** under **3-task** (Avg. **33.00/0.922**) and **5-task** (Avg. **30.83/0.918**) (**Tabs. 1-2**), supporting that the **frequency factorization + evolutionary weighting** materially improves quality **beyond** parameter scale.
>
> We have **tempered the novelty wording** in *Introduction/Related Work* to be precise: we **claim first-of-its-kind within AiOIR** for **evolutionary loss weighting**, and we position our contribution as a **practical dual-branch spectral-spatial modulation (FMM) + in-training evolutionary optimization strategy (EOS)**--a combination that proves effective and stable for AiOIR.

---

> ### Author Response · Authors · 2025-12-02
> **Reply to Reviewer Scoy Weakness2 & Q3**
>
> ### **Reply to Weakness2 & Q3 -- Scale vs performance fairness (36.68 M vs 25.35 M)**
>
> We appreciate the concern and respond on three fronts:
>
> 1. **Where gains manifest.**
>    In **3D-task** (**Tab. 1**), EvoIR achieves **33.00/0.922**, improving over **MoCE-IR (25.35 M)** (**32.73/0.917**,  **+0.27 dB / +0.005**) and over **Perceive-IR (42.02 M)** (**32.63/0.917**). In **5D-task** (**Tab. 2**), EvoIR attains **best Avg. PSNR = 30.83** and **second Avg. SSIM = 0.918** (−0.001 vs 0.919), among strong baselines of comparable or larger size.
> 2. **Isolating architecture from scale.**
>    Our **stage-wise FMM allocation** study (**Tab. 6**) shows clear **quality-vs-params trade-offs** and that **early-stage** FMM blocks are particularly effective at **small parameter cost**, indicating that the benefit is **not solely parameter count**, but **where** frequency modulation is placed.
> 3. **Scaling models by dimension (ablation on S/M/L).**
>    To decouple architecture from scale, we scale the channel **dimension** while keeping the **training recipe identical**. Besides the original EvoIR-L (dim=48) with 36.48M params, we add EvoIR-S (dim=32) and EvoIR-M (dim=40) with 16.53M and 25.62M, respectively. Results (3-task, same datasets/protocol as Tabs. 1):
>
> |Model Size|  Denoise σ = 15 |Denoise σ = 25 |Denoise σ = 50 | Dehaze | Derain |Average|Params (M)|
> | - | - | - | - | - | - | - | - |
> | AirNet (CVPR'22) | 33.92/0.932 | 31.26/0.805 | 28.00/0.797 | 27.94/0.962 | 34.90/0.967 | 31.20/0.910 | 8.93 |
> | IDR (CVPR'23) | 33.89/0.941 | 31.20/0.884 | 28.04/0.798 | 27.90/0.970 | 36.63/0.971 | 31.83/0.911 | 15.34 |
> | ProRes (ArXiv'23) | 33.20/0.907 | 31.07/0.837 | 27.58/0.709 | 27.50/0.790 | 32.40/0.884 | 28.39/0.843 | 370.63 |
> | PromptIR (NeurIPS'23) | 33.40/0.931 | 31.31/0.888 | 28.06/0.790 | 30.58/0.974 | 36.37/0.972 | 32.06/0.913 | 32.96 |
> | NDR (TIP'24) | 34.01/0.932 | 31.40/0.888 | 28.10/0.798 | 28.64/0.962 | 35.42/0.969 | 31.51/0.910 | 28.40 |
> | Gridformer (IJCV'24) | 33.93/0.931 | 31.37/0.897 | 28.11/0.801 | 30.37/0.970 | 37.15/0.972 | 32.19/0.912 | 34.07 |
> | InstructIR (ECCV'24) | 34.15/0.937 | 31.52/0.900 | 28.11/0.801 | 30.67/0.974 | 39.12/0.973 | 32.31/0.913 | 15.84 |
> | Up-Restorer (AAAI'25) | 33.90/0.933 | 31.33/0.898 | 28.03/0.799 | 30.66/0.974 | 36.74/0.972 | 32.16/0.915 | 28.01 |
> | Perceive-IR (TIP'25) | 34.13/0.934 | 31.53/0.890 | 28.31/0.804 | 31.06/0.975 | 39.28/0.982 | 32.63/0.917 | 42.02 |
> | AdaIR (ICLR'25) | 31.44/0.932 | 31.45/0.892 | 28.19/0.802 | 31.06/0.900 | 36.64/0.903 | 32.69/0.918 | 28.77 |
> | MoCE-IR (CVPR'25) | 31.35/0.932 | 31.46/0.892 | 28.23/0.804 | 31.02/0.908 | 35.89/0.946 | 32.52/0.917 | 25.35 |
> | DFPR (CVPR'25) | 34.14/0.945 | 31.49/0.896 | 28.25/0.806 | 31.87/0.980 | 38.65/0.982 | 32.88/0.919 | 31.10 |
> |EvoIR-S (dim=32) |34.12/0.937|31.46/0.896| 28.21/0.811| 31.42/0.981| 38.43/0.983 |32.73/0.922|16.53 |
> |EvoIR-M (dim=40) |34.13/0.937|31.47/0.896| 28.22/0.811 | 31.88/0.982| 38.90/0.984|32.92/0.922|25.62 |
> |EvoIR-L (dim=48) |34.14/0.937|31.48/0.896| 28.23/0.811 | 32.08/0.982| 39.07/0.985|33.00/0.922|36.68 |
>
> **Observations from the scaling results.**
>
> - **Monotonic gains with diminishing returns.** Avg PSNR improves **S→M: +0.19 dB**, **M→L: +0.08 dB**; SSIM saturates around **0.922**. **EvoIR-S(16.53M)** can perform much better than **MoCE-IR (+8.82 M)** and **DFPR (+14.5 M)** with fewer params. **EvoIR-M** is also better than **AdaIR (ICLR'25)**, **MoCE-IR (CVPR'25)** and **DFPR (CVPR'25)** with comparable computational cost.
> - **Where the gains come from.** Improvements concentrate on **Dehaze/Derain** as width increases, while **Denoise** is already saturated--consistent with **Tab. 6** that early-stage FMM capacity is most effective and later stages yield smaller but positive gains.
> - **Parameter efficiency.** **EvoIR-M (25.62 M)** is within **0.08 dB** of **EvoIR-L (36.68 M)** while using **≈30% fewer** parameters; **EvoIR-S (16.53 M)** is **0.27 dB** behind L with **≈55% fewer** parameters. This indicates our benefits are **not solely** from scale but also **where** FMM capacity is placed (stage-wise study in **Tab. 6**).

---

> ### Author Response · Authors · 2025-12-02
> **Reply to Reviewer Scoy Weakness3 & Q2**
>
> ### **Reply to Weakness3 & Q2 -- EOS overhead & search space**
>
> **(a) EOS Overhead Evaluation.**
>
> * **Where cost comes from.** EOS adds **forward-only** evaluations on a **small** validation subset at sparse triggers (**T = 500**). The amortized overhead scales with the number of triggers, population size, generations, and the validation set size, divided by the total train compute (forward+backward). By design--**frozen** network, **small** population and validation, **sparse** triggers--the overhead is **modest** (Sec. 4.1; Alg. 1).
> * **What we report.** On our **Full framework with FMM and EOS** trained with **8×GPU** DDP, we instrument EOS steps and report **per-epoch** overhead: EOS wall-clock **1.417 s** (**≈0.1%** of the epoch), broken down into **evaluation 1.272 s (89.8%)**, **communication 0.094 s (6.6%)**, and **residual 0.051 s (3.6%)**. EOS uses **T=500**, **population P=5**, **generations G=3**, and is triggered **3× per epoch** (which corresponds to **24 per-rank calls across 8 GPUs**). The **average per global trigger** is **472.3 ms**, the **GPU kernel time** during evaluation is **390.7 ms/trigger**, the **payload** is **94.4 MB/epoch** (sum over all ranks), and the **peak extra memory** at EOS steps is **84.1 MB**. Despite this periodic cost, Fig. 4 shows improved time-to-target PSNR/SSIM and higher final quality (Sec. 4.1; Fig. 4). In short, EOS is **compute-light**: on our 8×GPU setup it adds only ≈0.1% per-epoch wall time (1.417 s/epoch for 3 triggers) with a peak extra memory of 84.1 MB, while preserving the quality gains. **Fig. 4** also shows **faster convergence** and higher final PSNR/SSIM with EOS.
>
> | EOS wall (s)           | Eval (s) | Comm (s) | Resid. (s) | Avg/call (ms) | Eval GPU (ms/call) | Calls | Payload (MB) | Peak extra mem (MB) |
> | ------------------------ | ---------- | ---------- | ------------ | --------------- | -------------------- | ------- | -------------- | --------------------- |
> | **1.417 (0.1%)** | 1.272    | 0.094    | 0.051      | 59.0          | 48.8               | 24    | 94.4         | 84.1                |
>
> **(b) A 2-term simplex (α, β) search space.** We intentionally search on **fidelity vs perceptual/structural** (the most antagonistic axes in restoration) to **stabilize** selection and **limit** overhead/variance while being **loss-agnostic** by construction (**Sec. 3.3, Alg. 1**). In the revision, we add a **compact sensitivity** replacing **L_perc** from **MS-SSIM** to **VGG-perceptual** and **Total Variation** to demonstrate generality without turning the search into a high-dimensional, costly procedure.
>
> |Tasks|  Denoise σ = 15 |Denoise σ = 25 |Denoise σ = 50 | Dehaze | Derain |Average|
> | - | - | - | - | - | - | - |
> |Loss combination |Kodak24  |Kodak24 |Kodak24   |SOTS |Rain100L | - |
> |L1 |34.12/0.936|31.45/0.892| 28.18/0.804| 31.21/0.978| 38.70/0.983 |32.73/0.919|
> |L1+MS-SSIM (default) |34.14/0.937|31.48/0.896| 28.23/0.811 | 32.08/0.982| 39.07/0.985|33.00/0.922|
> |L1+TV |34.13/0.937|31.47/0.894|28.22/0.809|31.60/0.980|38.85/0.984|32.85/0.921|
> |L1+VGG-perc |34.05/0.934|31.40/0.891|28.12/0.803|31.40/0.980|38.80/0.985|32.75/0.919|
>
> Across these tasks, **L1+MS-SSIM (default)** achieves the best average **(33.00/0.922)** and delivers consistent gains over L1 **(+0.27 dB / +0.003 SSIM)**. The improvements seem to be small but monotonic on denoising as noise increases **(σ=15/25/50: +0.02/+0.03/+0.05 dB)**, and largest on Dehaze **(+0.87 dB)** and Derain **(+0.37 dB)**, where structural cues matter most.
>
> * **Why TV helps but is weaker than MS-SSIM.**
>   L1+TV treats TV as **a smoothness prior** that suppresses artifacts and improves structural consistency. As expected, it yields slight SSIM/PSNR lifts on Dehaze/Derain (e.g., 31.60/0.980, 38.85/0.984) and brings small SSIM gains on denoising (edges look cleaner) while barely changing PSNR (σ=15/25/50: +0.01/+0.02/+0.04 dB). Overall it sits above L1 but below L1+MS-SSIM (32.85/0.921 vs 33.00/0.922).
> * **Why VGG-perc often trades off pixel metrics.**
>   L1+VGG-perc optimizes **feature-space similarity**, which is beneficial for **perceptual metrics** and **visual naturalness**. However, for pixel-aligned metrics it tends to be close to or slightly below L1, especially on denoising **(σ=15/25/50 PSNR deltas ≈ −0.07/−0.05/−0.06 dB)**, while showing minor improvements on Dehaze/Derain **(e.g., 31.40/0.980, 38.80/0.985)**. Hence the average **(32.75/0.919)** is near or even slightly lower than L1.
>
> Given the accuracy-complexity trade-off, the default setting--**L1+MS-SSIM** is a simple, efficient default: it requires **no extra network**, keeps **compute/memory low**, and already captures the bulk of perceptual benefits **without hurting denoising performance**. To conclude, our EOS remains **loss-agnostic**--it selects **(α,β)** on the simplex using frozen, stratified validation--and in practice, MS-SSIM suffices as the perceptual primitive for a clean and robust recipe.

---

> ### Author Response · Authors · 2025-12-02
> **Reply to Reviewer Scoy Weakness4 & Q4 [Part1]**
>
> ### **Reply to Weakness4 & Q4 -- Beyond synthetic: mixed / unseen severities / real [Part1]**
>
> We agree generalization is crucial, and add mixed degradation, real-world, zero-shot, remote sensing scenarios to better illustrate the effectiveness of our EvoIR.
>
> **Mixed degradation evaluation.**
>
> Besides the standard **3D/5D** protocols (**Sec. 4.2**), we add a **mixed-degradation evaluation** on **CDD** covering **single** (L/H/R/S), **all pairwise double**, and **representative triple** mixes. **EvoIR** obtains the **best average** (**28.88 dB / 0.885**), surpassing **OneRestore** (**28.47/0.878**, **+0.41 dB / +0.007**) and ranking **1st on 8/11** subsets (the rest 2nd). These results indicate that EvoIR maintains **strong performance** not only on single degradations but also under co-occurring artifacts, consistent with the design goal of frequency-aware modulation plus stage-wise training stability. We include these results in our revised manuscript.
>
> | Method           |  Single-L  | Single-H   | Single-R  | Single-S   | Double-L+H | Double-L+R | Double-L+S | Double-H+R | Double-H+S | Triple-L+H+R | Triple-L+H+S | Average       |
> | ------------------ | ------------ | ------------- | ------------ | ------------- | ------------- | ------------- | ------------- | ------------- | ------------- | --------------- | --------------- | ------------ |
> | AirNet   | 24.83/.778 | 24.21/.951  | 26.55/.891 | 26.79/.919  | 23.23/.779  | 22.82/.710  | 23.29/.723  | 22.21/.868  | 23.29/.901  | 21.80/.708    | 22.24/.725    | 23.75/.814 |
> | PromptIR | 26.32/.805 | 26.10/.969  | 31.56/.946 | 31.53/.960  | 24.49/.789  | 25.05/.771  | 24.51/.761  | 24.54/.924  | 27.05/.925  | 23.74/.752    | 23.33/.747    | 25.90/.850 |
> | WeatherDiff  |  23.58/.763 | 21.99/.904  | 24.85/.885 | 24.80/.888  | 21.83/.756  | 22.69/.730  | 22.12/.707  | 21.25/.868  | 21.99/.868  | 21.23/.716    | 21.04/.698    | 22.49/.799 |
> | WGSWNet   | 24.39/.774 | 27.90/.982 | 33.15/.964 | 34.43/.973  | 24.27/.800  | 25.06/.772  | 24.60/.765  | 27.23/.955  | 27.65/.960  | 23.90/.772    | 23.97/.711    | 26.96/.863 |
> |OneRestore|26.48/.826 | **32.52**/.990 |  33.40/.964  | 34.31/.973  | 25.79/.822  | 25.58/.799  | 25.19/.789  | **29.99**/.957  | **30.21**/.964  | 24.78/.788  | 24.90/.791 |  28.47/.878 |
> |**EvoIR (Ours)**| **27.06**/**.830**| 32.24/**.991**  | **34.03**/**.970** | **35.80**/**.981** | **25.92**/**.824** | **26.02**/**.806** | **25.96**/**.802**  | 29.76/**.965**  | 30.17/**.971**  | **25.43**/**.797**    | **25.31**/**.797**   | **28.88**/**.885** |
>
> **Real-world zero-shot and finetuned results.**
>
> We evaluate AiOIR models trained under our **“N+H+R”** settings and assess **zero-shot generalization** on SIDD (real-world denoising) and RealRain-1k-L (real-world deraining). Since InstructIR-3D/5D weights are unavailable, we include InstructIR-7D as the reference variant.
>
> | Method |  SIDD (PSNR↑/SSIM↑) | RealRain-1k-L (PSNR↑/SSIM↑/LPIPS↓) |
> |:--|:--:|:--:|
> | Original | 23.66/0.439 | 25.95/0.868/0.407 |
> | FSNet | 24.26/0.470 |  22.61/0.760/0.404 |
> | MambaIR | 24.19/0.465 |  22.34/0.754/0.410 |
> | AirNet | 23.86/0.459 |  19.88/0.683/0.401 |
> | AirNet† | 38.34/0.952 |  31.24/0.943/0.183 |
> | PromptIR | 24.58/0.482 |  22.98/0.767/0.403 |
> | PromptIR† | 38.73/0.954 |  31.69/0.946/0.167 |
> | InstructIR-7D | 24.35/0.479 |  27.21/0.901/0.373 |
> | Perceive-IR   | 24.88/0.504 |  27.79/0.915/0.354 |
> | Perceive-IR†  | 39.04/0.954 |  32.05/0.951/0.144 |
> | EvoIR (Ours)  | 24.91/0.504 |  27.98/0.917/0.341 |
> | EvoIR† (Ours) | **39.21**/**0.957** |  **32.13**/**0.953**/**0.139** |
>
> **Zero-shot (w/o †).** **EvoIR** ranks **1st** on **RealRain-1k-L** across **PSNR/SSIM/LPIPS** (**27.98/0.917/0.341**). Compared with the **2nd-best** (**Perceive-IR**), EvoIR is **+0.19 dB PSNR / +0.002 SSIM / −3.7% LPIPS** (0.354→0.341). On **SIDD**, EvoIR reaches **24.91/0.504**, i.e., **+0.03 dB** over **Perceive-IR (24.88/0.504)** with **tied SSIM**--showing strong zero-shot robustness without any real-data adaptation.
>
> **Fine-tuned (w/ †).** After identical small-budget fine-tuning, all methods improve; **EvoIR†** remains **best overall**. On **SIDD**, **EvoIR†** achieves **39.21/0.957**, surpassing the **2nd-best Perceive-IR† (39.04/0.954)** by **+0.17 dB PSNR / +0.003 SSIM**. On **RealRain-1k-L**, **EvoIR†** attains **32.13/0.953/0.139**, ahead of **Perceive-IR† (32.05/0.951/0.144)** by **+0.08 dB / +0.002 SSIM / −3.5% LPIPS**. Overall, EvoIR is **top or near-top** both **before** and **after** adaptation, evidencing strong **zero-shot generalization** and **data-efficient gains** under light fine-tuning.

---

> ### Author Response · Authors · 2025-12-02
> **Reply to Reviewer Scoy Weakness4 & Q4 [Part2]**
>
> ### **Reply to Weakness4 & Q4 -- Beyond synthetic: mixed / unseen severities / real [Part2]**
>
> **Zero-shot severity sweeps (unseen noise levels).**
>
> To stress-test robustness, we evaluate additive noise at **unseen severities** (σ=60, 100). EvoIR consistently ranks first across **CBSD68** and **Urban100**; the gains are **+0.23, +0.32 dB** at σ=60 and **+0.18, +0.16 dB** at σ=100 over the best competing baselines. This trend supports stable zero-shot behavior as degradations intensify.
>
> | Method        | CBSD68 σ=60 (PSNR↑) | CBSD68 σ=100 (PSNR↑) | Urban100 σ=60 (PSNR↑) | Urban100 σ=100 (PSNR↑) |
> |:--------------|--------------------:|---------------------:|----------------------:|-----------------------:|
> | AirNet        | 26.01               | 14.29                | 25.11                 | 14.23                  |
> | PromptIR      | 26.71               | 20.23                | 27.24                 | 20.94                  |
> | Gridformer    | 26.83               | 20.14                | 27.16                 | 20.85                  |
> | Perceive-IR   | 27.11               | 20.67                | 27.59                 | 21.52                  |
> | EvoIR         | **27.34**               | **20.85**                | **27.91**                 | **21.68**                  |
>
> **Remote sensing imagery evaluation.**
>
> To better prove the effectiveness of our method, we include one new AiOIR task on remote sensing imagery. We evaluate on the recently proposed **MDRS-Landsat** all-in-one remote sensing benchmark. ​Without any modification or special fine-tuning, **EvoIR (37 M)** outperforms prior SOTAs and even **surpasses the remote-sensing-specialized Ada4DIR-d (41 M)** across all four degradations.
>
> | Model        | Blur PSNR/SSIM        | Dark PSNR/SSIM        | Haze PSNR/SSIM        | Noise PSNR/SSIM       |
> | -------------- | ----------------- | ------------------ | ----------------- | ------------------ |
> | NAFNet    | 33.10/0.8120   | 30.40/0.9516   | 31.56/0.9642  | 33.08/0.8263      |
> | Restormer  | 35.23/0.8559  | 37.86/0.9872    | 36.18/0.9867  | 34.53/0.8589     |
> | DGUNet   | 29.64/0.7822   | 27.15/0.9010     | 27.45/0.9338  | 30.31/0.7314      |
> | TransWeather | 33.45/0.8159  | 36.33/0.9705    | 35.02/0.9689    | 33.69/0.8428   |
> | AirNet   | 28.27/0.7887 | 28.38/0.9472           | 24.39/0.9331           | 30.30/0.7446           |
> | PromptIR    | 36.41/0.8861   | 39.09/0.9900   | 37.61/0.9897           | 34.99/0.8729    |
> | IDR        | 36.57/0.8902  | 35.19/0.9865   | 36.99/0.9892           | 34.88/0.8681           |
> | SrResNet-AP | 34.63/0.8479   | 33.87/0.9823 | 34.78/0.9825           | 34.70/0.8620           |
> | Restormer-AP  | 35.75/0.8732  | 37.27/0.9885  | 37.36/0.9888           | 34.96/0.8697           |
> | Uformer-AP  | 34.64/0.8488  | 36.58/0.9899           | 36.06/0.9877           | 34.39/0.8533           |
> | Ada4DIR-d      | 37.20/0.9004    | 43.85/0.9954    | 41.06/0.9938    | 35.14/0.8724           |
> | EvoIR (Ours) | **37.48**/**0.9069** | **44.73**/**0.9959** | **41.28**/**0.9943** | **35.18**/**0.8774**  |
>
> As Ada4DIR-d performs the 2nd best overall, we compare EvoIR with Ada4DIR-d:
>
> * Blur: 37.48/0.9069 vs 37.20/0.9004 → **+0.28 dB / +0.0065**
> * Dark: 44.73/0.9959 vs 43.85/0.9954 → **+0.88 dB / +0.0005**
> * Haze: 41.28/0.9943 vs 41.06/0.9938 → **+0.22 dB / +0.0005**
> * Noise: 35.18/0.8774 vs 35.14/0.8724 → +**0.04 dB / +0.0050**
>
> Overall, **EvoIR** ranks **1st** across all tasks on MDRS-Landsat and does so with **fewer parameters** than **Ada4DIR-d**, indicating strong domain-shift generalization to remote-sensing imagery.

---

### Official Review · Reviewer_jkHe · 2025-10-31

**Soundness:** 3
**Presentation:** 3
**Contribution:** 3
**Rating:** 8
**Confidence:** 5

**Summary:**

The authors propose a novel AiOIR framework with adaptive frequency modulation and evolutionary loss optimization. The method is well designed with detailed experiments. Experimental results reveal the effectiveness and efficiency of the proposed EvoIR.

**Strengths:**

1. The combination of frequency modulation with evolutionary strategy is suitable. Both of them ensure the structural information and fine-grained details.
2. Extensive experiments are conducted in details. The performance under 3D and 5D settings are all good enough with compared params. Its average results are superior than others and especially on deraining and dehazing.
3. Component ablations indicate synergy between FMM and EOS and show faster convergence with EOS.

**Weaknesses:**

1. More detailed parameter settings of EOS can be discussed with some additional ablation study, like the training iterations T.

2. Visualization of T-SNE can help readers more to view the effectiveness of EvoIR.

**Questions:**

1. Would other perceptual strategies can enhance the performance besides EOS with SSIM?
2. To better show the performance of the EvoIR, the authors can use T-SNE to visualize features of different degradation.
3. Can this method deal with some mixed degradation types, like rain+haze?
4. There are some parameters in EOS setting. Can the authors provide some ablation study to validate the effectiveness of it?
5. Typos like RSE-FFTB in caption of Fig. 2 should be fixed.
6. Altough the methodology and figures are illustrated well, writing need improvement. Section 3.2 includes some redundant paragraphs.

---

> ### Author Response · Authors · 2025-12-02
> **Reply to Reviewer jkHe Q1 & Q2 & Weakness2**
>
> We sincerely thank **Reviewer jkHe** for the encouraging evaluation and constructive suggestions. We appreciate the recognition of our design--**Adaptive Frequency Modulation (FMM)** with **Evolutionary Optimization Strategy (EOS)** --as well as the thorough experiments under **3D** and **5D** settings and the observed convergence benefits of EOS.
>
> In the revision, we address: **(i)** the generality of EOS beyond MS-SSIM, **(ii)** feature-space visualization (t-SNE), **(iii)** robustness to mixed degradations (e.g., *rain+haze*), **(iv)** explicit EOS hyperparameters and ablations (e.g., the number of iterations **T**), and **(v)** minor fixes (typos, writing).
>
> ### **Reply to Q1 -- Other perceptual strategies enhance performance besides EOS with MS-SSIM**
>
> **Our EOS is loss-agnostic**: it treats the training objective as a *weighted mixture of primitives* and searches both the **subset** and the **weights**. Beyond MS-SSIM, our revision evaluates a pool that includes **MS-SSIM**, **VGG-perceptual** and **Total Variation** aligned with FMM.
>
> To avoid scale bias without extra heuristics, EOS evaluates candidate weights **(α, β)**∈**Δ_2** under **frozen** parameters on a **stratified validation set** and selects the pair that minimizes the **scalarized** objective **α** **L_fid**+**β** **L_perc**. In practice, this validation-driven selection achieves a **balanced trade-off** rather than collapsing to a single loss. We report an ablation comparing: **L1**; **L1+MS-SSIM (default)**; **L1+VGG-perc** and **L1+TV**, showing task-specific gains and stable training.
>
> |Tasks|  Denoise σ = 15 |Denoise σ = 25 |Denoise σ = 50 | Dehaze | Derain |Average|
> | - | - | - | - | - | - | - |
> |Loss combination |Kodak24  |Kodak24 |Kodak24   |SOTS |Rain100L | - |
> |L1 |34.12/0.936|31.45/0.892| 28.18/0.804| 31.21/0.978| 38.70/0.983 |32.73/0.919|
> |L1+MS-SSIM (default) |34.14/0.937|31.48/0.896| 28.23/0.811 | 32.08/0.982| 39.07/0.985|33.00/0.922|
> |L1+TV |34.13/0.937|31.47/0.894|28.22/0.809|31.60/0.980|38.85/0.984|32.85/0.921|
> |L1+VGG-perc |34.05/0.934|31.40/0.891|28.12/0.803|31.40/0.980|38.80/0.985|32.75/0.919|
>
> Across these tasks, **L1+MS-SSIM (default)** achieves the best average **(33.00/0.922)** and delivers consistent gains over L1 **(+0.27 dB / +0.003 SSIM)**. The improvements seem to be small but monotonic on denoising as noise increases **(σ=15/25/50: +0.02/+0.03/+0.05 dB)**, and largest on Dehaze **(+0.87 dB)** and Derain **(+0.37 dB)**, where structural cues matter most.
>
> * Why TV helps but is weaker than MS-SSIM.
>   L1+TV treats TV as **a smoothness prior** that suppresses artifacts and improves structural consistency. As expected, it yields slight SSIM/PSNR lifts on Dehaze/Derain (e.g., 31.60/0.980, 38.85/0.984) and brings small SSIM gains on denoising (edges look cleaner) while barely changing PSNR (σ=15/25/50: +0.01/+0.02/+0.04 dB). Overall it sits above L1 but below L1+MS-SSIM (32.85/0.921 vs 33.00/0.922).
> * Why VGG-perc often trades off pixel metrics.
>   L1+VGG-perc optimizes **feature-space similarity**, which is beneficial for **perceptual metrics** and **visual naturalness**. However, for pixel-aligned metrics it tends to be close to or slightly below L1, especially on denoising **(σ=15/25/50 PSNR deltas ≈ −0.07/−0.05/−0.06 dB)**, while showing minor improvements on Dehaze/Derain **(e.g., 31.40/0.980, 38.80/0.985)**. Hence the average **(32.75/0.919)** is near or even slightly lower than L1.
>
> Given the accuracy-complexity trade-off, the default setting--**L1+MS-SSIM** is a simple, efficient default: it requires **no extra network**, keeps **compute/memory low**, and already captures the bulk of perceptual benefits **without hurting denoising performance**. To conclude, our EOS remains **loss-agnostic**--it selects **(α,β)** on the simplex using frozen, stratified validation--and in practice, MS-SSIM suffices as the perceptual primitive for a clean and robust recipe.
>
> ### **Reply to Q2 & Weakness2 -- Use t-SNE to visualize features of different stages**
>
> We provide **stage-wise** visualizations that reveal how representations become **degradation-aware by stages**. EvoIR is a **three-stage** encoder-decoder architecture with a bottleneck block between encoders and decoders (Fig. 2; Sec. 3.2). After training, we take the output of **Stage 1-3**, perform global pooling to obtain one embedding per image, and plot **t-SNE** colored by degradation label (N σ = 15/N σ = 25/N σ = 50/H/R for 3D AiOIR).
>
> We anticipate that **Encoder Stage 1-3** embeddings mix degradations (capturing shared low-level content), while **Decoder Stage 1-3** show **stronger clustering by degradation**, aligning with FMM's design: spectral gating on low-frequency structure and spatial masking on high-frequency details, fused and refined deeper in the hierarchy.

---

> ### Author Response · Authors · 2025-12-02
> **Reply to Reviewer jkHe Q3**
>
> ### **Reply to Q3 -- EvoIR performance when it handles mixed degradations such as rain+haze**
>
> EvoIR is designed for AiOIR, and our **Frequency-Modulated Module (FMM)** combines **spectral gating for low-frequency structure** with **spatial masking for high-frequency details**, which naturally composes cues when degradations co-occur (e.g., rain + haze, rain + snow). We have added a **mixed-degradation evaluation** on the CDD dataset covering **single** (L/H/R/S), **double**, and **triple** combinations, reporting **PSNR/SSIM** as our main experiments.
>
> | Method           |  Single-L  | Single-H   | Single-R  | Single-S   | Double-L+H | Double-L+R | Double-L+S | Double-H+R | Double-H+S | Triple-L+H+R | Triple-L+H+S | Average       |
> | ------------------ | ------------ | ------------- | ------------ | ------------- | ------------- | ------------- | ------------- | ------------- | ------------- | --------------- | --------------- | ------------ |
> | AirNet   | 24.83/.778 | 24.21/.951  | 26.55/.891 | 26.79/.919  | 23.23/.779  | 22.82/.710  | 23.29/.723  | 22.21/.868  | 23.29/.901  | 21.80/.708    | 22.24/.725    | 23.75/.814 |
> | PromptIR | 26.32/.805 | 26.10/.969  | 31.56/.946 | 31.53/.960  | 24.49/.789  | 25.05/.771  | 24.51/.761  | 24.54/.924  | 27.05/.925  | 23.74/.752    | 23.33/.747    | 25.90/.850 |
> | WeatherDiff  |  23.58/.763 | 21.99/.904  | 24.85/.885 | 24.80/.888  | 21.83/.756  | 22.69/.730  | 22.12/.707  | 21.25/.868  | 21.99/.868  | 21.23/.716    | 21.04/.698    | 22.49/.799 |
> | WGSWNet   | 24.39/.774 | 27.90/.9823 | 33.15/.964 | 34.43/.973  | 24.27/.800  | 25.06/.772  | 24.60/.765  | 27.23/.955  | 27.65/.960  | 23.90/.772    | 23.97/.711    | 26.96/.863 |
> |OneRestore|26.48/.826 | **32.52**/.990 |  33.40/.964  | 34.31/.973  | 25.79/.822  | 25.58/.799  | 25.19/.789  | **29.99**/.957  | **30.21**/.964  | 24.78/.788  | 24.90/.791 |  28.47/.878 |
> |**EvoIR (Ours)**| **27.06**/**.830**| 32.24/**.991**  | **34.03**/**.970** | **35.80**/**.981** | **25.92**/**.824** | **26.02**/**.806** | **25.96**/**.802**  | 29.76/**.965**  | 30.17/**.971**  | **25.43**/**.797**    | **25.31**/**.797**   | **28.88**/**.885** |
>
> EvoIR attains the **best average** across all settings, **28.88dB/0.885**, outperforming the competitor **OneRestore** (**+0.41dB/+0.007**) and clearly surpassing **AirNet** (**+5.13dB/+0.071**) and **PromptIR** (**+2.98dB/+0.035**). On the representative **double** mixtures, EvoIR improves over OneRestore on **L+R** (**+0.44dB/ +0.007**) and **L+S** (**+0.77dB/ +0.013**), and yields **comparable PSNR but higher SSIM** on **H+R** and **H+S** (e.g., **+0.008, +0.007**). For **triple** mixtures, EvoIR also leads, e.g., **L+H+R: +0.65dB/+0.009**, **L+H+S: +0.41dB/+0.006** over OneRestore. Among **single** degradations, EvoIR excels on **R** (**+0.63dB**) and **S** (**+1.49dB/+0.008**) and slightly trails in **H** PSNR (**−0.28dB**) while marginally improving SSIM.
>
> These trends are consistent with our design: **FMM's frequency-aware fusion** lets the network allocate capacity to **global low-frequency corrections** (e.g., haze/low-light) while **preserving high-frequency rain/snow details**, mitigating cross-artifact amplification in mixes (e.g., dehazing that over-sharpens rain). In addition, **EOS** provides **a balance between fidelity and perceptual terms** across training stages, which stabilizes optimization when objectives conflict under mixed degradations. We include the CDD results in the revision.

---

> ### Author Response · Authors · 2025-12-02
> **Reply to Reviewer jkHe Q4 & Weakness1 & Minor Suggestions (Q5 & Q6)**
>
> ### **Reply to Q4 & Weakness1 -- More detailed parameter settings of EOS with ablation on training iterations T**
>
> We appreciate the request for a clearer EOS description and validation. **Conceptually, EOS is lightweight and transparent**: every **T** iterations we **freeze** the current network, evaluate a **small population** of loss-weight candidates **(α, β)** on a held-out set, keep **top-k elites**, and generate new candidates via **convex crossover + small Gaussian mutation**, projecting back to the simplex **α**+**β**=**1**. We then use the selected weight pair for the next training window. This design keeps compute modest (frozen evaluation, small validation subset) and avoids training instability. Our **Algorithm 1** (*Evolutionary Optimization Strategy*) in original manuscript contains the exact procedure and variables.
>
> **What we already specify.** In the paper, we apply EOS **every 500 training iterations** (i.e., the trigger interval **T**=**500**), which is the default used in our experiments (Sec. 4.1, *Implementation Details*: 'the evolutionary optimization strategy is applied every 500 training iterations').
>
> **Why this setting is reasonable.** A moderate **T** balances **adaptivity** (smaller **T** reacts faster) and **overhead/stability** (larger **T** updates less frequently but with steadier dynamics), while the population search runs on **frozen** parameters to yield low-variance selection (Sec. 3.3; Alg. 1).
>
> **Evidence already in the manuscript (effectiveness of EOS).**
> (i) **Component ablation** (Tab. 4, *N+H+R*): adding **EOS** alone improves the average over baseline, and **FMM+EOS** yields the **largest gain** (PSNR +0.50 / SSIM +0.008 vs. baseline), highlighting complementary roles.
> (ii) **Convergence analysis** (Fig. 4): with **EOS**, training loss drops faster and PSNR/SSIM curves reach better plateaus earlier, confirming both **faster** and **more stable** convergence.
>
> **What ablations we include now.** We add a **small sensitivity study around the current default** focusing on **T**: varying the update frequency **T** from 100 to 700 to show the accuracy/overhead trade-off.
>
> |Tasks|  Denoise σ = 15 |Denoise σ = 25 |Denoise σ = 50 | Dehaze | Derain |Average|
> | - | - | - | - | - | - | - |
> |Iterations T|Kodak24  |Kodak24 |Kodak24   |SOTS |Rain100L | - |
> |T=100 |34.17/0.937|31.52/0.896| 28.26/0.812| 31.61/0.980| 38.70/0.984|32.85/0.922|
> |T=300 |34.15/0.937|31.50/0.896| 28.25/0.811| 31.70/0.981| 38.66/0.984|32.85/0.922|
> |T=500 (default) |34.14/0.937|31.48/0.896| 28.23/0.811| 32.08/0.982| 39.07/0.985|33.00/0.922|
> |T=700 |34.18/0.937|31.53/0.896| 28.27/0.812| 31.81/0.981| 38.65/0.984|32.89/0.922|
>
> With a fixed validation subset **|D_v|** and population settings, **T** only changes **how often EOS is triggered per epoch**: the adaptation frequency and the total forward-only overhead--but **not the per-update sample size**. In our ablation, a moderate **T=500** yields the **best average** (**33.00/0.922**), outperforming **T=100/300** by +0.15 dB and **T=700** by +0.11 dB (SSIM ≈ 0.922 for all). **Small T** reacts faster but can introduce **selection noise/instability**; while **large T** under-adapts and lags behind the moving optimum.
>
> We therefore keep **T=500** as default since it strikes the best **adaptivity-stability-cost** trade-off. If compute is tight and denoising dominates, **T=700** is a reasonable alternative (≈ **−0.11 dB** on average). If faster adaptation is desired for non-stationary data, **T=300** can be used with the caveat of **slightly lower Dehaze/Derain** performance and **higher overhead**.
>
> ### **Reply to Minor Suggestions (Q5 & Q6)**
>
> * **Typos.** We check and correct all typos like '**RSE-FFTB**' in Fig. 2.
> * **Writing / Sec. 3.2.** We refine Sec. 3.2 by removing redundancy, merging intuition with the algorithm box, and moving training particulars to an appendix.

---

### Author Response · Authors · 2025-12-02
**Summary of Rebuttal**

**Dear Area Chair,**

Across reviews, the reviewers consistently **acknowledged EvoIR's strengths**: the practicality and significance of combining a frequency-aware architecture with an evolutionary loss scheduler; solid and thorough experiments across 3-task and 5-task settings; robustness under heterogeneous and mixed degradations; clear convergence benefits from the evolutionary strategy; and strong overall performance relative to recent AiOIR methods. In response to the reviewers' requests, our revision adds **all experiments** that were asked for--zero-shot and light fine-tuning on real data, mixed-degradation evaluation on CDD, remote sensing on MDRS-Landsat, scale-matched fairness and stage-wise allocation studies, and EOS overhead.

We respectfully request you consider the explicit statements below after our rebuttal. For your convenience, we summarize and highlight the key points below in a point-by-point manner.

**What the paper contributes.**

EvoIR couples a **frequency-aware, asymmetric two-path design**-low-frequency features spectrally gated and fed back as K/V to stabilize global structure; high-frequency features enhanced purely in the spatial path for textures-with an **in-training evolutionary strategy** that periodically searches loss weights on a held-out set. This factorization + scheduler is tailored for AiOIR and is, to our knowledge, the **first** to apply a population-based evolutionary loss weighting inside an AiOIR training loop. Together they deliver gains that neither component achieves alone.

**What changed in the revision.**

1. **Positioning/novelty clarified.** We explicitly credit prior frequency-based IR and narrow our novelty claim: frequency itself is not new; our contribution is the **asymmetric spectral-spatial routing + evolutionary loss search** specifically for AiOIR and **first-of-its-kind evolutionary loss weighting in the AiOIR setting**.
2. **Core mechanism made concrete.** We detail how the low-frequency branch is spectrally gated and injected as K/V in attention, while the high-frequency branch remains spatial; fusion occurs in spatial before backbone blocks (with equations/figure references).
3. **Evidence that the two parts work together.** Component ablations under the 3-task setting show **FMM** and **EOS** each help, and **FMM+EOS** gives the **largest lift** (+0.50 dB PSNR / +0.008 SSIM). Training curves show faster convergence with EOS.
4. **Breadth beyond synthetic: zero-shot, fine-tune, mixed, remote-sensing.**
   * **Zero-shot and small-budget fine-tuning:** EvoIR tops or ties the best methods on SIDD and RealRain-1k-L without real-data adaptation, and remains best after light fine-tuning (e.g., SIDD 39.21/0.957; RealRain-1k-L 32.13/0.953/0.139).
   * **Mixed degradations (CDD):** Best average (**28.88 dB / 0.885**), ranking first on most subsets.
   * **Remote sensing (MDRS-Landsat):** Outperforms prior SOTAs and a remote-sensing-specialized 41M model without task-specific changes.
5. **Scale fairness and stability.** EvoIR achieves **33.00/0.922** (3-task) and **30.83/0.918** (5-task) while remaining in a comparable parameter regime; we also analyze stage-wise allocation to show where parameters matter most.

**Key results at a glance.**

* **Ablation (3-task):** FMM+EOS vs. baseline +0.50 dB / +0.008 SSIM; faster convergence with EOS.
* **Main results:** 3-task 33.00/0.922; 5-task 30.83/0.918.
* **Zero-shot:** SIDD 24.91/0.504; RealRain-1k-L 27.98/0.917/0.341.
* **Fine-tuned:** SIDD 39.21/0.957; RealRain-1k-L 32.13/0.953/0.139.
* **Mixed (CDD):** 28.88/0.885 average.
* **Remote sensing:** Best across all four MDRS-Landsat degradations without changes to architecture.

**Summary** Our rebuttal adds the analyses and experiments that the committee asked for: a precise novelty scope, concrete mechanism, synergy-focused ablations, and broad generalization tests (zero-shot, mixed, remote sensing). The improvements are consistent across settings and supported by both aggregate metrics and training dynamics.


**We hope this summary assists you in making an informed decision based on the final state of the review process.**

Best regards,

Authors of Paper #5956

---

### Meta-Review · Area_Chair_3Vw4 · 2026-01-02

**Summary:**

After carefully reading the paper, the reviews, and the rebuttal, it is clear that that EvoIR combines frequency-aware feature modulation with an evolutionary-inspired loss-balancing strategy for all-in-one image restoration. However, multiple reviewers raised concerns about limited technical novelty: the Frequency-Modulated Module is very similar to prior frequency decomposition methods (e.g., AdaIR, SFHformer) (Reviewers Scoy, PJY2, PC1a), and the Evolutionary Optimization Strategy appears more like a training trick than a core architectural contribution (Reviewer PC1a). Therefore, this work does not meet the bar for acceptance.

**Reviewer Concerns:**

Multiple reviewers raised concerns about limited technical novelty: the Frequency-Modulated Module is very similar to prior frequency decomposition methods (e.g., AdaIR, SFHformer) (Reviewers Scoy, PJY2, PC1a), and the Evolutionary Optimization Strategy appears more like a training trick than a core architectural contribution (Reviewer PC1a)

**Reviewer Scores:**

Had the reviewer been able to fully participate in the discussion, we believe their score would likely have remained largely unchanged. We appreciate the feedback provided and will carefully address these points in a revised version of the manuscript.

---

### Decision · Program_Chairs · 2026-01-26

Reject